# Variational Distillation of Diffusion Policies into Mixture of Experts

**Hongyi Zhou** [*†]    **Denis Blessing**[‡]    **Ge Li**[‡]    **Onur Celik**[‡§]
**Xiaogang Jia**[†‡]    **Gerhard Neumann**[‡§]    **Rudolf Lioutikov**[†]
[†] Intuitive Robots Lab, Karlsruhe Institute of Technology
[‡] Autonomous Learning Robots, Karlsruhe Institute of Technology
[§] FZI Research Center for Information Technology

## Abstract

This work introduces Variational Diffusion Distillation (VDD), a novel method that distills denoising diffusion policies into Mixtures of Experts (MoE) through variational inference. Diffusion Models are the current state-of-the-art in generative modeling due to their exceptional ability to accurately learn and represent complex, multi-modal distributions. This ability allows Diffusion Models to replicate the inherent diversity in human behavior, making them the preferred models in behavior learning such as Learning from Human Demonstrations (LfD). However, diffusion models come with some drawbacks, including the intractability of likelihoods and long inference times due to their iterative sampling process. The inference times, in particular, pose a significant challenge to real-time applications such as robot control. In contrast, MoEs effectively address the aforementioned issues while retaining the ability to represent complex distributions but are notoriously difficult to train. VDD is the first method that distills pre-trained diffusion models into MoE models, and hence, combines the expressiveness of Diffusion Models with the benefits of Mixture Models. Specifically, VDD leverages a decompositional upper bound of the variational objective that allows the training of each expert separately, resulting in a robust optimization scheme for MoEs. VDD demonstrates across nine complex behavior learning tasks, that it is able to: i) accurately distill complex distributions learned by the diffusion model, ii) outperform existing state-of-the-art distillation methods, and iii) surpass conventional methods for training MoE. The code and videos are available at https://intuitive-robots.github.io/vdd-website.

## 1 Introduction

Diffusion models [1–4] have gained increasing attention with their great success in various domains such as realistic image generation [5–8]. More recently, diffusion models have shown promise in Learning from Human Demonstrations (LfDs) [9–13]. A particularly challenging aspect of LfD is the high variance and multi-modal data distribution resulting from the inherent diversity in human behavior [14]. Due to the ability to generalize and represent complex, multi-modal distributions, diffusion models are a particularly suitable class of policy representations for LfD. However, diffusion models suffer from several drawbacks such as *long inference time* and *intractable likelihood calculation*. Many diffusion steps are required for high-quality samples leading to a *long inference time*, limiting the use in real-time applications such as robot control, where decisions are needed at a high frequency. Moreover, important statistical properties such as *exact likelihoods* are not easily obtained for diffusion models, which poses a significant challenge for conducting post hoc

---

[*]Correspondence to `hongyi.zhou@kit.edu`

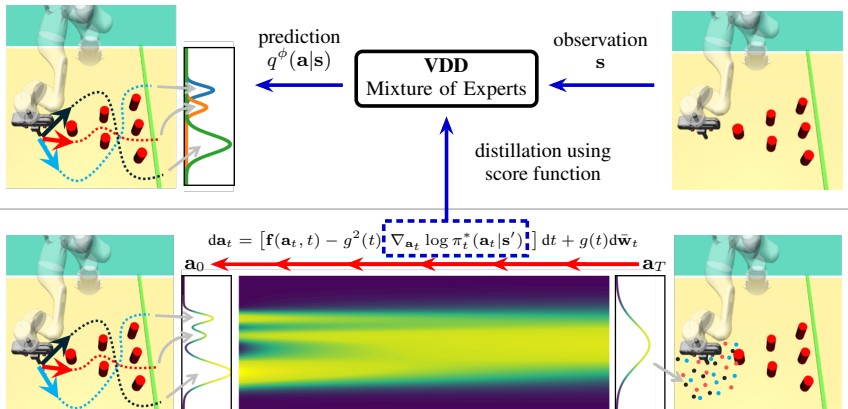

Figure 1: VDD distills a diffusion policy into an MoE. LfD is challenging due to the multimodality of human behaviour. For example, tele-operated demonstrations of an avoiding task often contain multiple solutions [13]. **Lower**: A diffusion policy can predict high quality actions but relies on an iterative sampling process from noise to data, shown as the red arrows. **Upper**: VDD uses the score function to distill a diffusion policy into an MoE, unifying the advantages of both approaches.

optimization such as fine-tuning through well-established reinforcement learning (RL) approaches like policy gradients or maximum entropy RL objectives.

A well-studied approach that effectively addresses these issues are Mixture of Experts (MoE). During inference, the MoE first selects an expert that is subsequently queried for a forward pass. This hierarchical structure provides a fast and simple sampling procedure, tractable likelihood computation, and the ability to represent multimodal distributions. These properties make them a well-suited policy representation for complex, multimodal behavior. However, training Mixture of Experts (MoEs) is often difficult and unstable [15]. The commonly used maximum likelihood objective can lead to undesired behavior due to mode-averaging, where the model fails to accurately represent certain modes. Yet, this limitation has been alleviated by recent methods that use alternative objectives, such as reverse KL-divergence, which do not exhibit mode-averaging behavior [14, 16].

To obtain the benefits of both models, i.e., learning highly accurate generative models with diffusion and obtaining simple, tractable models using a mixture of experts, this work introduces *Variational Diffusion Distillation* (VDD), a novel method that distills diffusion models to MoEs. Starting from the variational inference objective [17, 18], we derive a lower bound that decomposes the objective into separate per-expert objectives, resulting in a robust optimization scheme. Each per-expert objective elegantly leverages the gradient of the pre-trained score function such that the MoE benefits from the diffusion model's properties. The resulting MoE policy performs on par with the diffusion model and covers the same modes, while being interpretable, faster during inference, and has a tractable likelihood. This final policy is readily available to the user for post hoc analysis or fast fine-tuning for more specific situations. A high-level architecture of the VDD model and its relation to the diffusion policy is shown in Fig. 1. VDD is thoroughly evaluated on nine complex behavior-learning tasks that demonstrate the aforementioned properties. As an additional insight, this paper observed that one-step continuous diffusion models already perform well, a finding not discussed in prior work.

In summary, this work presents **VDD, a novel method** to distill diffusion models to MoEs, by **proposing a variational objective** leading to individual and robust expert updates, effectively leveraging a pre-trained diffusion model. The **thorough experimental evaluation** on nine sophisticated behavior learning tasks show that VDD *i) accurately distills* complex distributions, *ii) outperforms* existing SOTA distillation methods and *iii) surpasses* conventional MoE training methods.

## 2   Related Works

**Diffusion Models for Behavior Learning.** Diffusion models have been used in acquiring complex behaviors for solving sophisticated tasks in various learning frameworks. Most of these works train diffusion policies using offline reinforcement learning [19–24], or imitation learning [9, 12, 11, 10, 25]. In contrast, VDD distills diffusion models into an MoE policy to overcome diffusion-based

policy drawbacks such as *long inference times* or *intractable likelihoods* instead of optimizing policies directly from the data.

**Mixture of Experts (MoE) for Behavior Learning.** MoE models are well-studied, provide tractable likelihoods, and can represent multi-modality which makes them a popular choice in many domains such as in imitation learning [14, 26–31, 16, 13], reinforcement learning [32–38] and motion generation [39] to obtain complex behaviors. Although VDD also uses an MoE model, the behaviors are distilled from a pre-trained model using a variational objective and are not trained from scratch. The empirical evaluation demonstrates that VDD's *stable training procedure* results in improved performance compared to common MoE learning techniques.

**Knowledge Distillation from Diffusion Models** Knowledge distillation from diffusion models has been researched in various research areas. For instance, in text-to-3D modeling, training a NeRF-based text-to-3D model without any 3D data by mapping the 3D scene to a 2D image and leveraging a text-to-2D diffusion is proposed [7]. The work proposes minimizing the Score Distillation Sampling (SDS) loss that is inspired by probability density distillation [40] and incentivizes the 3D-model to be updated into higher density regions as indicated by the score function of the diffusion model. To overcome drawbacks such as over-smoothing and low-diversity problems when using the SDS loss, variational score distillation (VSD) treats the 3D scene as a random variable and optimizes a distribution over these scenes such that the projected 2D image aligns with the 2D diffusion model [8]. In a similar context, the work in [41] proposes distilling a trained diffusion model into another diffusion model while progressively reducing the number of steps. However, even though the number of diffusion steps is drastically reduced, a complete distillation, i.e. one-step inference as for VDD is not provided. Additionally, the resulting model suffers from the same drawbacks of diffusion models such as *intractable likelihoods*. In contrast, in consistency distillation (CD), diffusion models are distilled to consistency models (CM) [42–45] such that data generation is possible in one step from noise to data. However, one-step data generation typically results in lower sample quality, requiring a trade-off between iterative and single-step generation based on the desired outcome. As CMs, VDD performs one-step data generation but distills the pre-trained diffusion model to an MoE which has a *tractable likelihood* and is *efficient in inference time*. The experimental evaluations show the advantages of VDD over CMs. Diff-Instruct [46] proposes a two-step framework for distilling diffusion models into implicit generative models, whereas VDD considers an explicit generative model where the model's density can be directly evaluated. In addition, Diff-Instruct requires training an auxiliary diffusion model, while VDD only optimizes a single model. Score Regularized Policy Optimization (SRPO) [47] also leverages a diffusion behavior policy to regularize the offline RL-based objective. However, in contrast to SRPO, VDD learns an MoE policy instead of a uni-modal Gaussian policy and explicitly distills a diffusion model instead of using it as guidance during optimization. Furthermore, VDD trains MoEs policies in imitation learning instead of reward-labeled data as in offline RL. A concurrent work, EM-Distillation (EMD)[48], introduces an EM-style distillation objective derived from the mode-covering forward KL divergence. In contrast, VDD proposes an EM-style objective based on the mode-seeking reverse KL but encourages mode-covering behavior by having multiple experts.

## 3 Preliminaries

Here, we introduce the notation and foundation for Denoising Diffusion and Mixture of Experts policies. Throughout this work, we assume access to samples from a behavior policy $\pi^*$ and the corresponding state distribution $\mu$, that is $\mathbf{a} \sim \pi^*(\cdot|\mathbf{s})$ and $\mathbf{s} \sim \mu(\cdot)$, respectively.

**Denoising Diffusion Policies.** Denoising diffusion policies employ a diffusion process to smoothly convert data into noise. For a given state $\mathbf{s}'$, a diffusion process is modeled as stochastic differential equation (SDE) [3]

$$\mathrm{d}\mathbf{a}_t = \mathbf{f}(\mathbf{a}_t, t)\mathrm{d}t + g(t)\mathrm{d}\mathbf{w}_t, \quad \mathbf{a}_0 \sim \pi^*(\cdot|\mathbf{s}'), \quad \mathbf{s}' \sim \mu(\cdot) \tag{1}$$

with drift $\mathbf{f}$, diffusion coefficient $g(t)$ and Wiener process $\mathbf{w}_t \in \mathbb{R}^d$. The solution of the SDE is a diffusion process $(\mathbf{a}_t)_{t\in[0,T]}$ with marginal distributions $\pi_t^*$ such that $\pi_T \approx \mathcal{N}(\mathbf{0}, \mathbf{I})$ and $\pi_0 = \pi^*$. [49, 50] showed that the time-reversal of Eq. 1 is again an SDE given by

$$\mathrm{d}\mathbf{a}_t = \left[\mathbf{f}(\mathbf{a}_t, t) - g^2(t)\nabla_{\mathbf{a}_t} \log \pi_t^*(\mathbf{a}_t|\mathbf{s}')\right]\mathrm{d}t + g(t)\mathrm{d}\bar{\mathbf{w}}_t, \quad \mathbf{a}_T \sim \mathcal{N}(\cdot|\mathbf{0}, \mathbf{I}). \tag{2}$$

Simulating the SDE generates samples from $\pi^*(\cdot|\mathbf{s}')$ starting from pure noise. For most distributions $\pi^*$, however, we do not have access to the scores $(\nabla_{\mathbf{a}_t} \log \pi_t^*(\mathbf{a}_t|\mathbf{s}'))_{t\in[0,T]}$. The goal of diffusion-

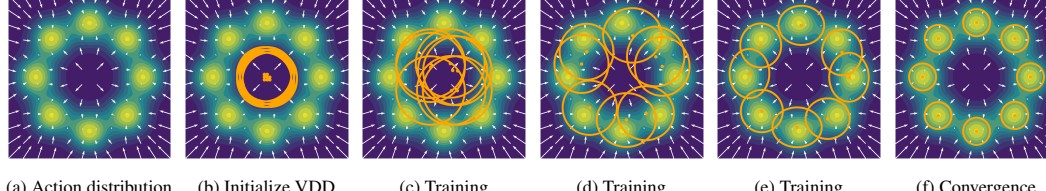

| (a) Action distribution | (b) Initialize VDD | (c) Training | (d) Training | (e) Training | (f) Convergence |

Figure 2: Illustration of training VDD using the score function for a fixed state in a 2D toy task. (a) The probability density of the distribution is depicted by the color map. The score function is shown by the gradient field, visualized as white arrows. From (b) to (f), we initialize and train VDD until convergence. We initialize 8 components, each represented by an orange circle. These components are driven by the score function to match the data distribution and avoid overlapping modes by utilizing the learning objective in Eq. (11). Eventually, they align with all data modes.

based modeling is therefore to approximate the intractable scores using a parameterized score function, i.e., $\boldsymbol{f}_\theta(\mathbf{a}, \mathbf{s}, t) \approx \nabla \log \pi_t^*(\mathbf{a}|\mathbf{s})$. To that end, several techniques have been proposed [51, 52], allowing for sample generation by approximately simulating Eq. 2. The most frequently employed SDEs in behavior learning are variance preserving (VP) [2, 10] and variance exploding (VE) [9]. For further details on diffusion-based generative modeling, we refer the reader to [3, 4]. While we only consider VE and VP in this work, VDD can be applied to any score-based method.

**Gaussian Mixtures of Expert Policies.** Mixtures of expert policies are conditional discrete latent variable models. Denoting the latent variable as $z$, the marginal likelihood can be decomposed as

$$q^\phi(\mathbf{a}|\mathbf{s}) = \sum_z q^\xi(z|\mathbf{s}) q^{\nu_z}(\mathbf{a}|\mathbf{s}, z), \tag{3}$$

where $q^\xi(z|\mathbf{s})$ and $q^{\nu_z}(\mathbf{a}|\mathbf{s}, z)$ are referred to as gating and experts respectively. $\xi$ and $\nu_z$ denote the gating and expert parameters and thus $\phi = \xi \cup \{\nu_z\}_z$. The gating is responsible for soft-partitioning the state space into sub-regions where the corresponding experts approximate the target density. To sample actions, that is, $\mathbf{a}' \sim q^\phi(\cdot|\mathbf{s}')$ for some state $\mathbf{s}'$, we first sample a component index from the gating, i.e., $z' \sim q^\xi(\cdot|\mathbf{s}')$. The component index selects the respective expert to obtain $\mathbf{a}' \sim q^{\nu_z}(\cdot|\mathbf{s}', z')$. For Gaussian MoE the experts are chosen as $q^{\nu_z}(\mathbf{a}|\mathbf{s}, z) = \mathcal{N}(\mathbf{a}|\mu^{\nu_z}(\mathbf{s}), \Sigma^{\nu_z}(\mathbf{s}))$, where $\mu^{\nu_z}, \Sigma^{\nu_z}$ could be neural networks parameterized by $\nu_z$. From the properties of Gaussian distributions, it directly follows that this model class admits tractable likelihoods and fast sampling routines. Furthermore, given enough components, Gaussian MoEs are universal approximators of densities [53], which makes them a good representation for distillation of diffusion policies.

## 4 Variational Distillation of Denoising Diffusion Policies

In this section, we outline the mathematical formulations of the VDD model. Detailed descriptions of the model architecture and algorithms can be found in Appendix A. We aim to distill a given diffusion policy $\pi(\mathbf{a}|\mathbf{s})$ by using a different policy representation $q^\phi$ with parameters $\phi$. This is useful, e.g., if $q^\phi(\mathbf{a}|\mathbf{s})$ has favorable properties such as likelihood tractability or admits fast inference schemes. Assuming that we can evaluate $\pi$ point-wise, a common approach is to leverage variational inference (VI) to frame this task as an optimization problem by minimizing the reverse Kullback-Leibler (KL) [54] divergence between $q^\phi$ and $\pi$, that is,

$$\min_\phi D_{\mathrm{KL}}(q^\phi(\mathbf{a}|\mathbf{s}') \| \pi(\mathbf{a}|\mathbf{s}')) = \min_\phi \mathbb{E}_{q^\phi(\mathbf{a}|\mathbf{s}')} \left[ \log q^\phi(\mathbf{a}|\mathbf{s}') - \log \pi(\mathbf{a}|\mathbf{s}') \right], \tag{4}$$

for a specific state $\mathbf{s}'$. To obtain a scalable optimization scheme, we combine amortized and stochastic VI [55]. The former allows for learning a conditional model $q^\phi(\mathbf{a}|\mathbf{s})$ instead of learning a separate $q^\phi$ for each state, while the latter allows for leveraging mini-batch computations, that is,

$$\min_\phi J(\phi) = \min_\phi \mathbb{E}_{\mu(\mathbf{s})} D_{\mathrm{KL}}(q^\phi(\mathbf{a}|\mathbf{s}) \| \pi(\mathbf{a}|\mathbf{s})) \approx \min_\phi \frac{M}{N} \sum_{\mathbf{s}_i \sim \mu} D_{\mathrm{KL}}(q^\phi(\mathbf{a}|\mathbf{s}_i) \| \pi(\mathbf{a}|\mathbf{s}_i)), \tag{5}$$

with batch size $M \le N$. Thus, $J(\phi)$ can be minimized using gradient-based optimization techniques with a gradient estimator such as reinforce [56, 57] or the reparameterization trick [58]. Note that, while the states are sampled from the given data set of the behavior policy, the actions needed to

evaluate the KL are generated using our estimated model $q^\phi(\mathbf{a}|\mathbf{s}_i)$. Yet, there are two difficulties to directly apply this scheme in distilling a diffusion model into an MoE: i) we are not able to evaluate the likelihood $\pi(\mathbf{a}|\mathbf{s})$ of a diffusion model, ii) training of MoE models is notoriously difficult [15]. We will address these two issues in Section 4.1 and Section 4.2, respectively.

## 4.1 Scalable Variational Inference for Denoising Diffusion Policy Distillation

Although we cannot directly evaluate the likelihood of the diffusion policy $\pi(\mathbf{a}|\mathbf{s})$, we have access to its score functions $\nabla_{\mathbf{a}_t} \log \pi_t(\mathbf{a}_t|\mathbf{s}) = \boldsymbol{f}_\theta(\mathbf{a}_t, \mathbf{s}, t)$, where $t \in [0, T]$ is the diffusion time step. In practice, we would like to evaluate the score in the limit of $t \to 0$ as $\nabla_\mathbf{a} \log \pi^*(\mathbf{a}|\mathbf{s}) \approx \lim_{t \to 0} \boldsymbol{f}_\theta(\mathbf{a}_t, \mathbf{s}, t)$. Yet, this might lead to an unstable optimization [59] as this score is often not estimated well throughout the action space, and, hence other diffusion time-step selection processes are needed [47]. For now, we will omit the diffusion time-step for the sake of simplicity and refer to Section 4.3 for a detailed discussion about time-step selection. Moreover, we will, for now, assume that the parametrization of $q^\phi$ is amendable to the reparameterization trick [58]. In this case, we can express $\mathbf{a} \sim q^\phi(\cdot|\mathbf{s})$ using a transformation $\boldsymbol{h}^\phi(\boldsymbol{\epsilon}, \mathbf{s})$ with an auxiliary variable $\boldsymbol{\epsilon} \sim p(\cdot)$ such that $\mathbf{a} = \boldsymbol{h}^\phi(\boldsymbol{\epsilon}, \mathbf{s})$. We then express the gradient of $J$ w.r.t. $\phi$ as

$$\nabla_\phi J(\phi) \approx \frac{M}{N} \sum_{\mathbf{s}_i \sim \mu} \mathbb{E}_{p(\boldsymbol{\epsilon})} \left[ \nabla_\phi \log q^\phi(\boldsymbol{h}^\phi(\boldsymbol{\epsilon}, \mathbf{s}_i)|\mathbf{s}_i) - \nabla_\phi \log \pi(\boldsymbol{h}^\phi(\boldsymbol{\epsilon}, \mathbf{s}_i)|\mathbf{s}_i) \right]. \quad (6)$$

Using the chain rule for derivatives, it is straightforward to see that

$$\nabla_\phi \log \pi(\mathbf{a}|\mathbf{s}_i) = (\nabla_\mathbf{a} \log \pi(\mathbf{a}|\mathbf{s}_i)) \nabla_\phi \boldsymbol{h}^\phi(\boldsymbol{\epsilon}, \mathbf{s}_i) = \boldsymbol{f}_\theta(\mathbf{a}, \mathbf{s}_i, t) \nabla_\phi \boldsymbol{h}^\phi(\boldsymbol{\epsilon}, \mathbf{s}_i). \quad (7)$$

As $\nabla_\mathbf{a} \log \pi(\mathbf{a}|\mathbf{s}_i)$ can be replaced by the given score of the pre-trained diffusion policy, we can directly use of VI for optimizing $J$ without evaluating the likelihoods of $\pi$.

## 4.2 Variational Inference via Mixture of Experts

To distill multimodal distributions learned by diffusion models, we require a more complex family of distributions than conditional diagonal Gaussian distributions, which are commonly used in amortized VI. We will therefore use Gaussian mixture of experts. To that end, we construct an upper bound of $J$ which is decomposable into single objectives per expert, allowing for reparameterizing each expert individually and therefore avoiding the need for techniques that perform reparameterization for the entire MoE [60, 61]. The upper bound $U(\phi, \tilde{q})$ can be obtained by making use of the chain rule for KL divergences [62, 63, 15], i.e.,

$$J(\phi) = U(\phi, \tilde{q}) - \mathbb{E}_{\mu(\mathbf{s})} \mathbb{E}_{q^\phi(\mathbf{a}|\mathbf{s})} D_{\mathrm{KL}} \left( q^\phi(z|\mathbf{a}, \mathbf{s}) \| \tilde{q}(z|\mathbf{a}, \mathbf{s}) \right), \quad (8)$$

where $\tilde{q}$ is an arbitrary auxiliary distribution and upper bound $U(\phi, \tilde{q}) =$

$$\mathbb{E}_{\mu(\mathbf{s})} [\mathbb{E}_{q^\xi(z|\mathbf{s})} [\underbrace{\mathbb{E}_{q^{\nu_z}(\mathbf{a}|\mathbf{s}, z)} \left[ \log q^{\nu_z}(\mathbf{a}|\mathbf{s}, z) - \log \pi(\mathbf{a}|\mathbf{s}) - \log \tilde{q}(z|\mathbf{a}, \mathbf{s}) \right]}_{U_z^\mathbf{s}(\nu_z, \tilde{q})} + \log q^\xi(z|\mathbf{s})]], \quad (9)$$

with $U_z^\mathbf{s}(\nu_z, \tilde{q})$ being the objective function for a single expert $z$ and state $\mathbf{s}$. For further details see Appendix B. Since the expected KL term on the right side of Eq. 8 is always positive, it directly follows that $U$ is an upper bound on $J$ for any $\tilde{q}$. This gives rise to an optimization scheme similar to the expectation-maximization algorithm [64], where we alternate between minimization (M-Step) and tightening of the upper bound $U$ (E-Step), that is,

$$\min_\phi U(\phi, \tilde{q}) \qquad \text{and} \qquad \min_{\tilde{q}} \mathbb{E}_{\mu(\mathbf{s})} \mathbb{E}_{q^\phi(\mathbf{a}|\mathbf{s})} D_{\mathrm{KL}} \left( q^\phi(z|\mathbf{a}, \mathbf{s}) \| \tilde{q}(z|\mathbf{a}, \mathbf{s}) \right), \quad (10)$$

respectively. Please note that $\tilde{q}$ is fixed during the M-Step and $\phi$ during the E-Step. In what follows, we identify the M-Step as a hierarchical VI problem and elaborate on the E-Step.

**M-Step for Updating the Experts.** The decomposition in Eq. 8 allows for optimizing each expert separately. The optimization objective for a specific expert $z$ and state $\mathbf{s}$, that is, $U_z^\mathbf{s}(\nu_z, \tilde{q})$, is

$$\min_{\nu_z} U_z^\mathbf{s}(\nu_z, \tilde{q}) = \min_{\nu_z} \mathbb{E}_{q^{\nu_z}(\mathbf{a}|\mathbf{s}, z)} \left[ q^{\nu_z}(\mathbf{a}|\mathbf{s}, z) - \log \pi(\mathbf{a}|\mathbf{s}) - \log \tilde{q}(z|\mathbf{a}, \mathbf{s}) \right]. \quad (11)$$

Note that this objective corresponds to the standard reverse KL objective from variational inference (c.f. Eq. 4) , with an additional term $\log \tilde{q}(z|\mathbf{a}, \mathbf{s})$, which acts as a repulsion force, keeping the

individual components from concentrating on the same mode. Assuming that $\tilde{q}$ is differentiable and following the logic in Section 4.1 it is apparent that the single component objective in Eq 11 can be optimized only by having access to scores $\nabla_{\mathbf{a}} \log \pi(\mathbf{a}|\mathbf{s})$. Moreover, we can again leverage amortized stochastic VI, that is $\min_{\nu_z} \mathbb{E}_{\mu(\mathbf{s})} [U_z^{\mathbf{s}}(\nu_z, \tilde{q})] \approx \min_{\nu_z} \frac{M}{N} \sum_{\mathbf{s}_i \sim \mu} U_z^{\mathbf{s}_i}(\nu_z, \tilde{q})$.

**M-Step for Updating the Gating.** The M-Step for the gating parameters, i.e., minimizing $U(\phi, \tilde{q})$ with respect to $\xi \subset \phi$ is given by

$$\min_{\xi} U(\phi, \tilde{q}) = \max_{\xi} \mathbb{E}_{\mu(\mathbf{s})} \mathbb{E}_{q^{\xi}(z|\mathbf{s})} \left[ q^{\xi}(z|\mathbf{s}) - U_z^{\mathbf{s}}(\nu_z, \tilde{q}) \right]. \tag{12}$$

Using gradient estimators such as reinforce [57], requires evaluating $U_z^{\mathbf{s}}(\nu_z, \tilde{q})$ which is not possible as we do not have access to $\log \pi(\mathbf{a}|\mathbf{s})$. This motivates the need for a different optimization scheme. We note that $q^{\xi}$ is a categorical distribution and can approximate any distribution over $z$. It does therefore not suffer from problems associated with the forward KL divergence such as mode averaging due to limited complexity of $q^{\xi}$. We thus propose using the following objective for optimizing $\xi$, i.e.,

$$\min_{\xi} \mathbb{E}_{\mu(\mathbf{s})} D_{\mathrm{KL}}(\pi(\mathbf{a}|\mathbf{s}) \| q^{\phi}(\mathbf{a}|\mathbf{s})) = \max_{\xi} \mathbb{E}_{\mu(\mathbf{s})} \mathbb{E}_{\pi(\mathbf{a}|\mathbf{s})} \mathbb{E}_{\tilde{q}(z|\mathbf{a},\mathbf{s})} \left[ \log q^{\xi}(z|\mathbf{s}) \right], \tag{13}$$

resulting in a cross-entropy loss that does not require evaluating the intractable $\log \pi(\mathbf{a}|\mathbf{s})$. Moreover, we note that using the forward KL does not change the minimizer as

$$\xi^* = \arg\min_{\xi} \mathbb{E}_{\mu(\mathbf{s})} D_{\mathrm{KL}}(\pi(\mathbf{a}|\mathbf{s}) \| q^{\phi}(\mathbf{a}|\mathbf{s})) = \arg\min_{\xi} \mathbb{E}_{\mu(\mathbf{s})} D_{\mathrm{KL}}(q^{\phi}(\mathbf{a}|\mathbf{s}) \| \pi(\mathbf{a}|\mathbf{s})), \tag{14}$$

and, therefore does not affect the convergence guarantees of our method. Please note that the forward KL requires samples from the teacher model $\pi$. In practice, if the dataset used for training the diffusion model is available, it can be used as a proxy and prevent costly data regeneration.

**E-Step: Tighening the Lower Bound.** Using the properties of the KL divergence, it can easily be seen that the global minimizer of the E-Step, i.e., the optimization objective defined in Eq. 10 can be found by leveraging Bayes' rule, i.e.,

$$\tilde{q}(z|\mathbf{a}, \mathbf{s}) = q^{\phi^{\mathrm{old}}}(z|\mathbf{a}, \mathbf{s}) = \frac{q^{\nu_z^{\mathrm{old}}}(\mathbf{a}|\mathbf{s}, z) q^{\xi^{\mathrm{old}}}(z|\mathbf{s})}{\sum_z q^{\nu_z^{\mathrm{old}}}(\mathbf{a}|\mathbf{s}, z) q^{\xi^{\mathrm{old}}}(z|\mathbf{s})}. \tag{15}$$

The superscript 'old' refers to the previous iteration. Numerically, $\phi^{\mathrm{old}}$ can easily be obtained by using a stop-gradient operation which is crucial as $\tilde{q}$ is fixed during the subsequent M-step which requires blocking the gradients of $\tilde{q}$ with respect to $\phi$. As the KL is set to zero after this update, the upper bound is tight after every E-step ensuring $\mathbb{E}_{\mu(\mathbf{s})} D_{\mathrm{KL}}(q^{\phi}(\mathbf{a}|\mathbf{s}) \| \pi^{\theta}(\mathbf{a}|\mathbf{s})) = U(\phi, \tilde{q})$. Hence, VDD has similar convergence guarantees to EM, i.e., every update step improves the original objective.

### 4.3 Choosing the Diffusion-Timestep

In denoising diffusion models, the score function is usually characterized as a time-dependent function $\nabla_{\mathbf{a}_t} \log \pi_t(\mathbf{a}_t|\mathbf{s}) = \boldsymbol{f}_{\theta}(\mathbf{a}_t, \mathbf{s}, t)$, where $t$ is the diffusion timestep. Yet, the formulation in Section 4.1 only leverages the pretrained diffusion model at time $t \to 0$. However, [47]showed that using an ensemble of scores from multiple diffusion time steps significantly improves performances. We, therefore, replace Eq. 11 with a surrogate objective that utilizes scores at different time steps, that is,

$$\min_{\nu_z} U_z^{\mathbf{s}}(\nu_z, \tilde{q}) = \min_{\nu_z} \mathbb{E}_{q^{\nu_z}(\mathbf{a}|\mathbf{s}, z)} \mathbb{E}_{p(t)} \left[ q^{\nu_z}(\mathbf{a}|\mathbf{s}, z) - \log \pi(\mathbf{a}|\mathbf{s}, t) - \log \tilde{q}(z|\mathbf{a}, \mathbf{s}) \right], \tag{16}$$

with $p(t)$ being a distribution on $[0, T]$. Furthermore, we provide an ablation study for different time step selection schemes and empirically confirm the findings from [47].

## 5 Experiments

We conducted imitation learning experiments by distilling two types of diffusion models: variance preserving (**VP**) [2, 12] and variance exploding (**VE**) [65, 4]. We selected **DDPM** as the representative for VP and **BESO** as the representative for VE. We adopt the choices of samplers and the number of denoising steps in [9] and [13]. Additional evaluation of teacher models with different numbers of denoising steps can be found in Appendix F. In the experiments, **VP-1** and **VE-1** denote the

| | VP (DDPM) | VE (BESO) | VP-1 | VE-1 | CD-VE | CTM-VE | VDD-VP(ours) | VDD-VE(ours) |
|---|---|---|---|---|---|---|---|---|
| **Kitchen** | 3.35 | 4.06 | 0.22 | 4.02 | 3.87 ± 0.05 | **3.89 ± 0.11** | 3.24 ± 0.12 | 3.85 ± 0.10 |
| **Block Push** | 0.96 | 0.96 | 0.09 | 0.94 | 0.89 ± 0.05 | 0.89 ± 0.04 | 0.93 ± 0.03 | **0.91 ± 0.03** |
| **Avoiding** | 0.94 | 0.96 | 0.09 | 0.84 | 0.82 ± 0.05 | 0.93 ± 0.02 | 0.92 ± 0.02 | **0.95 ± 0.01** |
| **Aligning** | 0.85 | 0.85 | 0.00 | 0.93 | **0.94 ± 0.08** | 0.81 ± 0.11 | 0.70 ± 0.07 | 0.86 ± 0.04 |
| **Pushing** | 0.74 | 0.78 | 0.00 | 0.70 | 0.66 ± 0.05 | 0.80 ± 0.07 | 0.61 ± 0.04 | **0.85 ± 0.02** |
| **Stacking-1** | 0.89 | 0.91 | 0.00 | 0.75 | 0.69 ± 0.06 | 0.54 ± 0.17 | 0.81 ± 0.08 | **0.85 ± 0.02** |
| **Stacking-2** | 0.68 | 0.70 | 0.00 | 0.53 | 0.46 ± 0.11 | 0.30 ± 0.09 | **0.60 ± 0.07** | 0.57 ± 0.06 |
| **Sorting (Image)** | 0.69 | 0.70 | 0.20 | 0.68 | 0.71 ± 0.07 | 0.70 ± 0.07 | **0.80 ± 0.04** | 0.76 ± 0.04 |
| **Stacking (Image)** | 0.58 | 0.66 | 0.00 | 0.58 | 0.63 ± 0.01 | 0.59 ± 0.10 | **0.78 ± 0.02** | 0.60 ± 0.04 |

(a) Task Success Rate (or Environment Return for Kitchen)

| | VP (DDPM) | VE (BESO) | VP-1 | VE-1 | CD-VE | CTM-VE | VDD-VP(ours) | VDD-VE(ours) |
|---|---|---|---|---|---|---|---|---|
| **Avoiding** | 0.89 | 0.87 | 0.25 | 0.76 | 0.72 ± 0.02 | **0.79 ± 0.04** | 0.37 ± 0.01 | 0.72 ± 0.12 |
| **Aligning** | 0.62 | 0.67 | 0.00 | 0.34 | 0.32 ± 0.14 | 0.31 ± 0.28 | 0.25 ± 0.09 | **0.40 ± 0.04** |
| **Pushing** | 0.74 | 0.76 | 0.00 | 0.50 | 0.53 ± 0.07 | 0.54 ± 0.08 | 0.66 ± 0.05 | **0.69 ± 0.08** |
| **Stacking-1** | 0.24 | 0.30 | 0.00 | 0.26 | **0.19 ± 0.12** | 0.18 ± 0.08 | 0.19 ± 0.05 | 0.16 ± 0.03 |
| **Stacking-2** | 0.12 | 0.13 | 0.00 | 0.07 | 0.03 ± 0.05 | 0.09 ± 0.06 | 0.07 ± 0.04 | **0.13 ± 0.06** |
| **Sorting (Image)** | 0.16 | 0.19 | 0.09 | 0.14 | 0.14 ± 0.06 | 0.08 ± 0.05 | 0.12 ± 0.03 | **0.22 ± 0.03** |
| **Stacking (Image)** | 0.31 | 0.15 | 0.00 | 0.10 | 0.06 ± 0.01 | 0.04 ± 0.04 | 0.05 ± 0.02 | **0.11 ± 0.03** |

(b) Task Entropy

Table 1: Comparison of distillation performance, (a) VDD achieves on-par performance with Consistency Distillation (CD) (b) VDD is able to possess versatile skills (indicated by high task entropy) while keeping high success rate. The best results for distillation are bolded, and the highest values except origin models are underlined. In most tasks VDD achieves both high success rate and entropy. **Note:** to better compare the distillation performance, we report the performance of origin diffusion model, therefore only seed 0 results of diffusion models are presented here.

results when performing only one denoising step of the respective diffusion models during inference. **VDD-VP** and **VDD-VE** denote the results of distilled VDD Additionally, we consider the SoTA Consistency Distillation (**CD**) [42] and Consistency Trajectory Model (**CTM**) [44] as baselines for comparing VDD's performance in distillation. For CD and CTM, we distill from the VE following the original works. For CTM we adapt the implementation and design choices from Consistency Policy [45], which are specialized for behavior learning. We compare VDD against MoE learning baselines, namely the widely-used Expectation-Maximization (**EM**) [66] approach as a representative of the maximum likelihood-based objective and the recently introduced SoTA method Information Maximizing Curriculum (**IMC**) as representative of the reverse KL-based objective. To make them stronger baselines, we extend them with the architecture described in Figure 5 and name the extended methods as **EM-GPT** and **IMC-GPT**, respectively. For a fair comparison, we used the same diffusion models as the origin model for all distillation methods that we have trained on seed 0. For a statistically significant comparison, all methods have been run on *4 random seeds*, and the mean and the standard deviation are reported throughout the evaluation. Detailed descriptions regarding the baselines implementation and hyperparameters selection can be found in Appendix D and E.

The evaluations are structured as follows. Firstly, we demonstrate that VDD is able to achieve competitive performance with the SoTA diffusion distillation method and the original diffusion models on two established datasets. Next, we proceed to a recently proposed challenging benchmark with human demonstrations, where VDD outperforms existing diffusion distillation and SoTA MoE learning approaches. We then highlight the faster inference time of VDD. Following this, a series of ablation studies reflect the importance of VDD's essential algorithmic properties. Finally, we provide a visualization to offer deeper insights into our method.

### 5.1 Competitive Distillation Performance in Imitation Learning Datasets

We first demonstrate the effectiveness of VDD using two widely recognized imitation learning datasets: Relay Kitchen [67] and XArm Block Push [68]. A detailed description of these environments is provided in Appendix C. To ensure a fair comparison, we follow the same evaluation process as outlined in [9]. The environment rewards for Relay Kitchen and the success rate for XArm Block Push are presented in Table 1a with mean and standard deviation resulting from 100 environment rollouts. The results indicate that VDD achieves a performance comparable to CD in both tasks, with slightly better outcomes in the block push dataset. An additional interesting finding is that BESO, with only one denoising step (VE-1), already proves to be a strong baseline in these tasks, as the original models outperformed the distillation results in both cases. We attribute this interesting

| Environments | EM-GPT | IMC-GPT | VDD-VP | VDD-VE | EM-GPT | IMC-GPT | VDD-VP | VDD-VE |
|---|---|---|---|---|---|---|---|---|
| Avoiding | $0.65 \pm 0.18$ | $0.75 \pm 0.08$ | $0.92 \pm 0.02$ | $\mathbf{0.95 \pm 0.01}$ | $0.17 \pm 0.13$ | $\mathbf{0.82 \pm 0.05}$ | $0.37 \pm 0.01$ | $0.73 \pm 0.09$ |
| Aligning | $0.78 \pm 0.04$ | $0.83 \pm 0.02$ | $0.70 \pm 0.07$ | $\mathbf{0.86 \pm 0.04}$ | $0.38 \pm 0.11$ | $0.27 \pm 0.09$ | $0.25 \pm 0.09$ | $\mathbf{0.40 \pm 0.04}$ |
| Pushing | $0.16 \pm 0.07$ | $0.76 \pm 0.04$ | $0.61 \pm 0.04$ | $\mathbf{0.85 \pm 0.02}$ | $0.14 \pm 0.10$ | $0.31 \pm 0.03$ | $0.66 \pm 0.05$ | $\mathbf{0.69 \pm 0.08}$ |
| Stacking-1 | $0.58 \pm 0.06$ | $0.54 \pm 0.05$ | $0.81 \pm 0.08$ | $\mathbf{0.83 \pm 0.09}$ | $\mathbf{0.43 \pm 0.08}$ | $0.37 \pm 0.04$ | $0.19 \pm 0.05$ | $0.16 \pm 0.03$ |
| Stacking-2 | $0.34 \pm 0.07$ | $0.29 \pm 0.07$ | $\mathbf{0.60 \pm 0.07}$ | $0.57 \pm 0.06$ | $\mathbf{0.27 \pm 0.05}$ | $0.17 \pm 0.07$ | $0.07 \pm 0.04$ | $0.13 \pm 0.06$ |
| Sorting (image) | $0.69 \pm 0.02$ | $0.74 \pm 0.04$ | $\mathbf{0.80 \pm 0.04}$ | $0.76 \pm 0.03$ | $0.13 \pm 0.03$ | $0.10 \pm 0.03$ | $0.12 \pm 0.03$ | $\mathbf{0.22 \pm 0.03}$ |
| Stacking (image) | $0.04 \pm 0.03$ | $0.39 \pm 0.10$ | $\mathbf{0.78 \pm 0.02}$ | $0.60 \pm 0.04$ | $0.00 \pm 0.00$ | $0.05 \pm 0.04$ | $0.08 \pm 0.02$ | $\mathbf{0.11 \pm 0.03}$ |
| Relay Kitchen | $3.62 \pm 0.10$ | $3.67 \pm 0.05$ | $3.24 \pm 0.12$ | $\mathbf{3.85 \pm 0.10}$ | - | - | - | - |
| Block Push | $0.88 \pm 0.04$ | $0.89 \pm 0.04$ | $\mathbf{0.93 \pm 0.03}$ | $0.91 \pm 0.03$ | - | - | - | - |

Table 2: Comparison between VDD and SoTA MoE approaches, with **left: success rate** and **right: entropy**. VDD consistently outperforms EM and IMC in terms of task success. For behavior versatility, VDD outperforms in 4 out of 7 D3IL tasks.

| NFE | 1 | 8 | 32 | 64 |
|---|---|---|---|---|
| VE (BESO) | 4.03 | 10.38 | 32.14 | 60.64 |
| VP (DDPM) | **2.15** | 8.25 | 29.47 | 55.62 |
| **VDD (Ours)** | **2.16** | | | |

| NFE | 1 | 4 | 8 | 16 |
|---|---|---|---|---|
| VE (BESO) | 9.69 | 12.72 | 16.39 | 23.68 |
| VP (DDPM) | 6.49 | 9.33 | 12.79 | 19.90 |
| **VDD (Ours)** | **4.91** | | | |

Table 3: Inference time in state-based pushing (**left**) and image-based stacking (**right**). The gray shaded area indicates the default setting for diffusion models.

observation to the possibility that the Relay Kitchen and the XArm Block Push tasks are comparably easy to solve and do not provide diverse, multi-modal data distributions. We therefore additionally evaluate the methods on a more recently published dataset (D3IL) [13] which is explicitly generated for complex robot imitation learning tasks and provides task entropy measurements.

## 5.2 Replicating Diffusion Performance while Possessing Versatile Behavior

The D3IL benchmark provides human demonstrations for several challenging robot manipulation tasks, focusing on evaluating methods in terms of both success rate and versatility, i.e. the different behaviors that solve the same task. This benchmark includes a versatility measure for each task, referred to as **task entropy**. Task entropy is a scalar value ranging from 0 to 1, where 0 indicates the model has only learned one way to solve the task, and 1 indicates the model has covered all the skills demonstrated by humans. Detailed descriptions of the environments and the calculation of task entropy are provided in the Appendix C. The task success rate of the distilled polices is presented in Table 1a. The results show VDD outperforms consistency distillation and 1-step variants of origin models in 6 out of 7 tasks except the Aligning. However, in the Aligning task VDD achieves higher task entropy, indicating more diverse learned behaviors. The task entropy is presented in Table 1b. The results demonstrate that VDD achieves higher task entropy compared to both consistency models (CD, CTM) and the 1-step diffusion models (VP-1, VE-1) in 4 out of 7 tasks, which shows that our method replicates high-quality versatile behaviors from diffusion policies.

## 5.3 Comparison with MoE learning from scratch

The previous evaluation demonstrated that VDD can effectively distill diffusion models into MoEs, preserving the performance and behavioral versatility. In this section, we discuss the necessity of using VDD instead of directly learning MoEs from scratch by comparing VDD against EM-GPT and IMC-GPT. Both methods train MoE models from scratch but differ in their objectives. While EM is based on the well-known maximum likelihood objective, IMC is based on a reverse KL objective. The results in Table 2 show that VDD consistently outperforms both, EM-GPT and IMC-GPT across a majority of all tasks in terms of both success rate and task entropy. We attribute the performance boost leveraging the generalization ability of diffusion models and the stable updates provided by our decomposed lower bound Eq.(11).

## 5.4 Fast Inference with distilled MoE

Inference with MoE models do not require an iterative denoising process and are therefore faster in sampling. We evaluate the inference time of VDD against DDPM and BESO on the state-based *pushing* task and the image-based *stacking* task and report the average results from 200 predictions in Table 3. In addition to the absolute inference time in milliseconds, we report the *number of function*

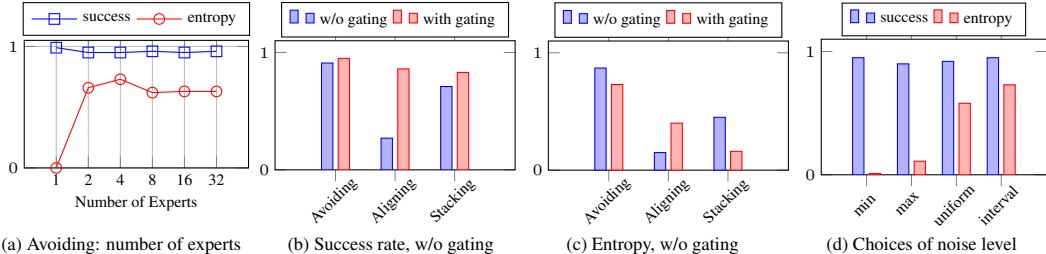

| (a) Avoiding: number of experts | (b) Success rate, w/o gating | (c) Entropy, w/o gating | (d) Choices of noise level |

Figure 3: Ablation studies for key design choices used in VDD. (a) Using only one expert leads to a higher success rate but is unable to solve the task in diverse manners. Sufficiently more experts can trade off task success and action diversities. (b)Learning the gating distribution improves the success rates in three D3IL tasks. (c) A Uniform gating leads to higher task entropy in three out of two tasks. (d) Sampling the score from multiple noise levels leads to a better distillation performance

*evaluations (NFE)* in Table 3 for better comparability. The results show that VDD is significantly faster than the original diffusion models in both cases, even when the diffusion model takes only one denoising step. For a fair comparison, all methods used an identical number of transformer layers. The predictions were conducted using the same system (RTX 3070 GPU, Intel i7-12700 CPU).

## 5.5  Ablation Studies

We assess the importance of VDD's key properties on different environments by reporting the task performance and task entropy averaged over four different seeds.

**Number of experts matters for task entropy.** We start by varying the number of experts of the MoE model while freezing all other hyperparameters on the *avoiding* task. Figure 3a shows the average task success rate and task entropy of MoE models trained with VDD. The success rate is almost constantly high for all numbers of experts, except for the single expert (i.e. a Gaussian policy) case which shows a slightly higher success rate. However, the single expert can only cover a single mode of the multi-modal behavior space and hence achieves a task entropy of 0. With an increasing number of experts, the task entropy increases and eventually converges after a small drop.

**Training a gating distribution boosts performance.** Figure 3b shows the success rates when training a parameterized gating network $q^\xi(z|s)$ (red) and when fixing the probability of choosing expert $z$ to $q(z) = 1/N$, where $N$ is the number of experts (blue). While training a gating distribution increases the success rate over three different tasks, the task entropy (see Figure 3c) slightly decreases in two out of three tasks. This observation makes sense as the MoE with a trained gating distribution leads to an input-dependent specialization of each expert, while the experts with a fixed gating are forced to solve the task in every possible input.

**Interval time step sampling increases task entropy.** Here, we explore different time step distributions $p(t)$, as introduced in Eq.(16). We consider several methods in Fig. 3d: using the minimum time step, i.e., $p(t) = \lim_{t \to 0} \delta(t)$, where $\delta$ denotes a Dirac delta distribution, the maximum time step $p(t) = \delta(T)$, a uniform distribution on $[0, T]$ and on sub-intervals $[t_0, t_1] \subset [0, T]$ with interval bounds $t_0, t_1$ being hyperparameters. While success rates were comparable across the variants, interval sampling yielded the highest task entropy with very high success rates. Thus, interval time-step sampling is adopted as our default setting. The results were obtained from the avoiding task.

## 5.6  Visualization of the per-Expert Behavior

We provide additional visualizations on the *Avoiding* task from the D3IL task suite aiming to provide further intuition on how VDD leverages the individual experts. Figure 4 illustrates the expert selection according to the likelihood of the gating distribution at a given state, offering several key insights. First, VDD effectively distills experts with distinct behaviors, e.g., $z_1$ typically moves downward, $z_2$ tends to move upward, while $z_3$ and $z_4$ tend to generate horizontal movements. Second, the gating mechanism effectively deactivates redundant experts ($z_6, z_7, z_8$) in most states, demonstrating that a larger number of components can be used without harming performance, as the gating mechanism deactivates redundant experts. Lastly, using a single component ($Z = 1$) can achieve a perfect success rate at the cost of losing behavior diversity. On the contrary, using many experts potentially results in

a slightly lower success rate but increased behavior diversity. These qualitative results are consistent with the quantitative results from the ablation study presented in Figure 3a.

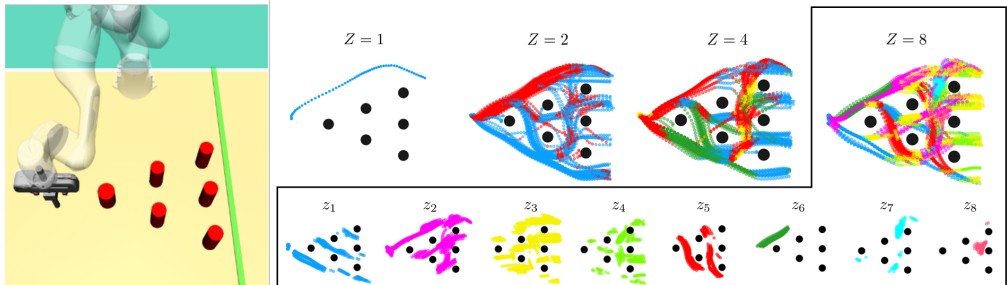

Figure 4: Trajectory visualization for VDD with different number of components $Z \in \{1, 2, 4, 8\}$ on the *Avoiding* task (left). Different colors indicate components with highest likelihood according to the learned gating network $q^\xi(z|\mathbf{s})$ at a state $\mathbf{s}$. For each step we select the action by first sampling an expert from the categorical gating distribution and then take the mean of the expert prediction. We decompose the case $Z = 8$ and visualize the individual experts $z_i$ (bottom row). Diverse behavior emerges as multiple actions are likely given the same state. For example, moving to the bottom right ($z_1$) and top right ($z_2$). An extreme case of losing diversity is seen with $Z = 1$, where the policy is unable to capture the diverse behavior of the diffusion teacher, leading to deterministic trajectories.

## 6 Conclusion

This work introduced Variational Diffusion Distillation (VDD), a novel method that distills a diffusion model to an MoE. VDD enables the MoE to benefit from the diffusion model's properties like generalization and complex, multi-modal data representation, while circumventing its shortcomings like long inference time and intractable likelihood calculation. Based on the variational objective, VDD derives a lower bound that enables optimizing each expert individually. The lower-bound leads to a stable optimization and elegantly leverages the gradient of the pre-trained score function such that the overall MoE model effectively benefits from the diffusion model's properties. The evaluations on nine sophisticated behavior learning tasks show that VDD achieves on-par or better distillation performance compared to SOTA methods while retaining the capability of learning versatile skills. The ablation on the number of experts reveals that a single expert is already performing well, but can not solve the tasks in a versatile manner. Additionally, the results show that training the gating distribution greatly boosts the performance of VDD, but reduces the task entropy.

**Limitations.** VDD is not straightforwardly applicable to generating very high-dimensional data like images due to the MoE's contextual mean and covariance prediction. Scaling VDD to images requires further extensions like prediction in a latent space. Additionally, the number of experts needs to be pre-defined by the user. However, a redundantly high number of experts could increase VDD's training time and potentially decrease the usage in post hoc fine-tuning using reinforcement learning. Similar to other distillation methods, the performance of VDD is bounded by the origin model.

**Future Work.** A promising avenue for further research is to utilize the features of the diffusion 'teacher' model to reduce training time and enhance performance. This can be achieved by leveraging the diffusion model as a backbone and fine-tuning an MoE head to predict the means and covariance matrices of the experts. The time-dependence of the diffusion model can be directly employed to train the MoE on multiple noise levels, effectively eliminating the need for the time-step selection scheme introduced in Section 4.3.

**Broader Impact.** Improving and enhancing imitation learning algorithms could make real-world applications like robotics more accessible, with both positive and negative impacts. We acknowledge that it falls on sovereign governments' responsibility to identify these potential negative impacts.

## Acknowledgement

We thank Moritz Reuss for the valuable discussions and technical support. H.Z. and R.L. acknowledges funding by the German Research Foundation (DFG) – 448648559. D.B. is supported by

funding from the pilot program Core Informatics of the Helmholtz Association (HGF). G.L. is supported in part by the Helmholtz Association of German Research Centers. G.N. was supported in part by Carl Zeiss Foundation through the Project JuBot (Jung Bleiben mit Robotern). The authors also acknowledge support by the state of Baden-Württemberg through HoreKa supercomputer funded by the Ministry of Science, Research and the Arts Baden-Württemberg and by the German Federal Ministry of Education and Research.

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

# A  VDD Architecture and Algorithm Box

**Algorithm 1** VDD training

**Require:** observations $S$, actions $A$, pretrained
denoising score function $s^\theta(s,a)$
1: Initialize MoE:
   $q^\phi(\mathbf{a}|\mathbf{s}) = \sum_z q^\xi(z|\mathbf{s})q^{\nu_z}(\mathbf{a}|\mathbf{s},z)$
2: **for** each iteration $i = 1, 2, \ldots, N$ **do**
3:   **M-Step: Update Experts:**
4:   update $q^{\nu_z}(\mathbf{a}|\mathbf{s},z)$ with Eq. 16
5:   **E-Step: Update Gating:**
6:   compute gating targets with Eq. 15
7:   update $q^\xi(z|\mathbf{s})$ using cross-entropy loss
8: **end for**
9: Output learned MoE $q^\phi(a|s)$

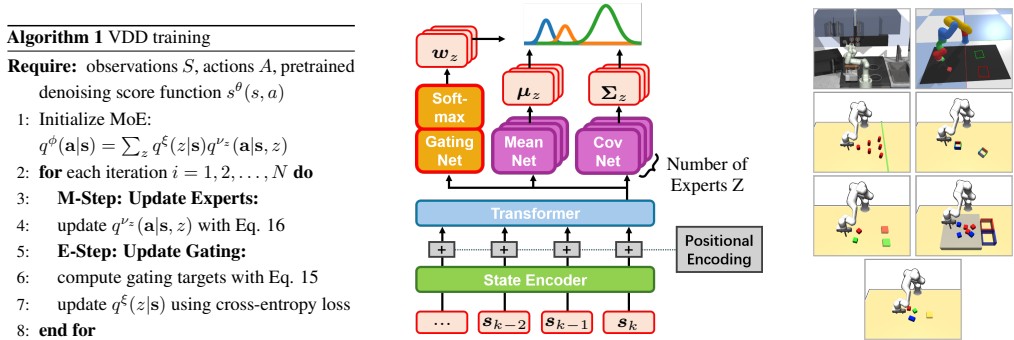

Figure 5: VDD's architecture    Figure 6: Task Envs.

The architecture of VDD is depicted in Fig. 5 and the algorithm box is given in Algorithm 1. Align with the SoTA diffusion policies' architecture, such as [9, 12], we leverage a transformer architecture for VDD to encode a short observation history, seen as a sequence of states $\{s_k\}$. The state sequences are first encoded using a state encoder: a linear layer for state-based tasks or a pre-trained ResNet-18 for image-based tasks. Positional encodings are added to the encoded sequence, which is then fed into a decoder-only transformer. The last output token predicts the parameters of the MoE, including the mean and covariance of each expert, as well as the gating distribution for expert selection. This parameterization enables VDD to predict all experts and the gating distribution with a single forward pass, enhancing inference time efficiency.

# B  Derivations

## B.1  Derivation of the Variational Decomposition (Eq. 8)

Recall that the marginal likelihood of the Mixture of Experts is given as

$$q^\phi(\mathbf{a}|\mathbf{s}) = \sum_z q^\xi(z|\mathbf{s})q^{\nu_z}(\mathbf{a}|\mathbf{s},z), \tag{17}$$

where $z$ denotes the latent variable, $q^\xi(z|\mathbf{s})$ and $q^{\nu_z}(\mathbf{a}|\mathbf{s},z)$ are referred to as gating and experts respectively.

The expected reverse Kullback-Leibler (KL) divergence between $q^\phi$ and $\pi$ is given as

$$\min_\phi \mathbb{E}_{\mu(\mathbf{s})} D_{\mathrm{KL}}(q^\phi(\mathbf{a}|\mathbf{s})\|\pi(\mathbf{a}|\mathbf{s})) = \min_\phi \mathbb{E}_{\mu(\mathbf{s})} \mathbb{E}_{q^\phi(\mathbf{a}|\mathbf{s})} \left[\log q^\phi(\mathbf{a}|\mathbf{s}) - \log \pi(\mathbf{a}|\mathbf{s})\right] \tag{18}$$

$$= \min_\phi J(\phi), \tag{19}$$

where

$$J(\phi) = \int_{\mathbf{s}} \mu(\mathbf{s}) \int_{\mathbf{a}} q^\phi(\mathbf{a}|\mathbf{s}) \left[\log q^\phi(\mathbf{a}|\mathbf{s}) - \log \pi(\mathbf{a}|\mathbf{s})\right] d\mathbf{a}d\mathbf{s}. \tag{20}$$

We can write

$$J(\phi) = \int_{\mathbf{s}} \mu(\mathbf{s}) \sum_z q^\xi(z|\mathbf{s}) \int_{\mathbf{a}} q^{\nu_z}(\mathbf{a}|\mathbf{s},z) \left[\log q^\phi(\mathbf{a}|\mathbf{s}) - \log \pi(\mathbf{a}|\mathbf{s})\right] d\mathbf{a}d\mathbf{s}, \tag{21}$$

where we have used the definition of the marginal likelihood of the Mixture of Experts in Eq. 3.

With the identity

$$q^\phi(\mathbf{a}|\mathbf{s}) = \frac{q^\xi(z|\mathbf{s})q^{\nu_z}(\mathbf{a}|\mathbf{s},z)}{q^\phi(z|\mathbf{s},\mathbf{a})} \tag{22}$$

we can further write

$$J(\phi) = \int_{\mathbf{s}} \mu(\mathbf{s}) \sum_z q^\xi(z|\mathbf{s}) \int_{\mathbf{a}} q^{\nu_z}(\mathbf{a}|\mathbf{s}, z) \left[ \log q^\xi(z|\mathbf{s}) + \log q^{\nu_z}(\mathbf{a}|\mathbf{s}, z) - \log q(z|\mathbf{s}, \mathbf{a}) \right.$$
$$\left. - \log \pi(\mathbf{a}|\mathbf{s}) \right] d\mathbf{a} d\mathbf{s}. \tag{23}$$

We can now introduce the auxiliary distribution $\tilde{q}(z|\mathbf{a}, \mathbf{s})$ by adding and subtracting it as

$$J(\phi) = \int_{\mathbf{s}} \mu(\mathbf{s}) \sum_z q^\xi(z|\mathbf{s}) \int_{\mathbf{a}} q^{\nu_z}(\mathbf{a}|\mathbf{s}, z) \left[ \log q^\xi(z|\mathbf{s}) + \log q^{\nu_z}(\mathbf{a}|\mathbf{s}, z) - \log q^\phi(z|\mathbf{s}, \mathbf{a}) \right.$$
$$\left. - \log \pi(\mathbf{a}|\mathbf{s}) + \log \tilde{q}(z|\mathbf{a}, \mathbf{s}) - \log \tilde{q}(z|\mathbf{a}, \mathbf{s}) \right] d\mathbf{a} d\mathbf{s}. \tag{24}$$

We can rearrange the terms such that

$$J(\phi) = \int_{\mathbf{s}} \mu(\mathbf{s}) \sum_z q^\xi(z|\mathbf{s}) \int_{\mathbf{a}} q^{\nu_z}(\mathbf{a}|\mathbf{s}, z) \left[ \log q^\xi(z|\mathbf{s}) + \log q^{\nu_z}(\mathbf{a}|\mathbf{s}, z) - \log \pi(\mathbf{a}|\mathbf{s}) \right.$$
$$\left. - \log \tilde{q}(z|\mathbf{a}, \mathbf{s}) \right] d\mathbf{a} d\mathbf{s} + \int_{\mathbf{s}} \mu(\mathbf{s}) \sum_z q^\xi(z|\mathbf{s}) \int_{\mathbf{a}} q^{\nu_z}(\mathbf{a}|\mathbf{s}, z) \left[ \log \tilde{q}(z|\mathbf{a}, \mathbf{s}) \right.$$
$$\left. - \log q^\phi(z|\mathbf{s}, \mathbf{a}) \right] d\mathbf{a} d\mathbf{s}. \tag{25}$$

With can plug in the identity

$$q^{\nu_z}(\mathbf{a}|\mathbf{s}, z) = \frac{q^\phi(\mathbf{a}|\mathbf{s}) q^\phi(z|\mathbf{s}, \mathbf{a})}{q^\xi(z|\mathbf{s})} \tag{26}$$

into the second sum and obtain

$$J(\phi) = \int_{\mathbf{s}} \mu(\mathbf{s}) \sum_z q^\xi(z|\mathbf{s}) \int_{\mathbf{a}} q^{\nu_z}(\mathbf{a}|\mathbf{s}, z) \left[ \log q^\xi(z|\mathbf{s}) + \log q^{\nu_z}(\mathbf{a}|\mathbf{s}, z) - \log \pi(\mathbf{a}|\mathbf{s}) - \log \tilde{q}(z|\mathbf{a}, \mathbf{s}) \right] d\mathbf{a} d\mathbf{s}$$
$$+ \int_{\mathbf{s}} \sum_z \int_{\mathbf{a}} q^\phi(\mathbf{a}|\mathbf{s}) q^\phi(z|\mathbf{s}, \mathbf{a}) \left[ \log \tilde{q}(z|\mathbf{a}, \mathbf{s}) - \log q^\phi(z|\mathbf{s}, \mathbf{a}) \right] d\mathbf{a} d\mathbf{s}. \tag{27}$$

which is the expected negative KL as

$$J(\phi) = \int_{\mathbf{s}} \mu(\mathbf{s}) \sum_z q^\xi(z|\mathbf{s}) \int_{\mathbf{a}} q^{\nu_z}(\mathbf{a}|\mathbf{s}, z) \left[ \log q^\xi(z|\mathbf{s}) + \log q^{\nu_z}(\mathbf{a}|\mathbf{s}, z) - \log \pi(\mathbf{a}|\mathbf{s}) - \log \tilde{q}(z|\mathbf{a}, \mathbf{s}) \right] d\mathbf{a} d\mathbf{s}$$
$$- \mathbb{E}_{\mu(\mathbf{s})} \mathbb{E}_{q^\phi(\mathbf{a}|\mathbf{s})} D_{\mathrm{KL}}(q^\phi(z|\mathbf{s}, \mathbf{a}) \| \tilde{q}(z|\mathbf{a}, \mathbf{s})). \tag{28}$$

We note that

$$U(\phi, \tilde{q}) = \int_{\mathbf{s}} \mu(\mathbf{s}) \sum_z q^\xi(z|\mathbf{s}) \int_{\mathbf{a}} q^{\nu_z}(\mathbf{a}|\mathbf{s}, z) \left[ \log q^\xi(z|\mathbf{s}) + \log q^{\nu_z}(\mathbf{a}|\mathbf{s}, z) - \log \pi(\mathbf{a}|\mathbf{s}) \right.$$
$$\left. - \log \tilde{q}(z|\mathbf{a}, \mathbf{s}) \right] d\mathbf{a} d\mathbf{s}, \tag{29}$$

such that we arrive to the identical expression as in Eq. 8

$$J(\phi) = U(\phi, \tilde{q}) - \mathbb{E}_{\mu(\mathbf{s})} \mathbb{E}_{q^\phi(\mathbf{a}|\mathbf{s})} D_{\mathrm{KL}} \left( q^\phi(z|\mathbf{a}, \mathbf{s}) \| \tilde{q}(z|\mathbf{a}, \mathbf{s}) \right). \tag{30}$$

## B.2   Derivation of the Gating Update (Eq. 13)

First, we note that

$$D_{\mathrm{KL}}(\pi(\mathbf{a}|\mathbf{s}) \| q^\phi(\mathbf{a}|\mathbf{s})) = \mathbb{E}_{\pi(\mathbf{a}|\mathbf{s})} \left[ \log \pi(\mathbf{a}|\mathbf{s}) \right] - \mathbb{E}_{\pi(\mathbf{a}|\mathbf{s})} \left[ \log q^\phi(\mathbf{a}|\mathbf{s}) \right] \tag{31}$$

as we are optimizing w.r.t. $\phi$, we can write $const. = \mathbb{E}_{\pi(\mathbf{a}|\mathbf{s})} \left[ \log \pi(\mathbf{a}|\mathbf{s}) \right]$

$$D_{\mathrm{KL}}(\pi(\mathbf{a}|\mathbf{s}) \| q^\phi(\mathbf{a}|\mathbf{s})) = -\mathbb{E}_{\pi(\mathbf{a}|\mathbf{s})} \left[ \log q^\phi(\mathbf{a}|\mathbf{s}) \right] + const. \tag{32}$$

We can now introduce the latent variable $z$

$$D_{\mathrm{KL}}(\pi(\mathbf{a}|\mathbf{s})\|q^\phi(\mathbf{a}|\mathbf{s})) = -\mathbb{E}_{\pi(\mathbf{a}|\mathbf{s})}\left[\sum_z \tilde{q}(z|\mathbf{a},\mathbf{s})\log q^\phi(\mathbf{a}|\mathbf{s})\right]$$
$$+ const. \tag{33}$$

We use the identity in Eq. 22 to arrive at

$$D_{\mathrm{KL}}(\pi(\mathbf{a}|\mathbf{s})\|q^\phi(\mathbf{a}|\mathbf{s})) = -\mathbb{E}_{\pi(\mathbf{a}|\mathbf{s})}\left[\sum_z \tilde{q}(z|\mathbf{a},\mathbf{s})\left(\log q^{\nu_z}(\mathbf{a}|\mathbf{s},z) + \log q^\xi(z|\mathbf{s}) - \log q^\phi(z|\mathbf{s},\mathbf{a})\right)\right]$$
$$\tag{34}$$

$$+ const.$$

$$= -\mathbb{E}_{\pi(\mathbf{a}|\mathbf{s})}\left[\sum_z \tilde{q}(z|\mathbf{a},\mathbf{s})\left(\log q^{\nu_z}(\mathbf{a}|\mathbf{s},z) + \log q^\xi(z|\mathbf{s}) - \log q^\phi(z|\mathbf{s},\mathbf{a})\right)\right.$$
$$\left. + \log \tilde{q}(z|\mathbf{a},\mathbf{s}) - \log \tilde{q}(z|\mathbf{a},\mathbf{s})\right] + const. \tag{35}$$

$$= -\mathbb{E}_{\pi(\mathbf{a}|\mathbf{s})}\left[\sum_z \tilde{q}(z|\mathbf{a},\mathbf{s})\left(\log q^{\nu_z}(\mathbf{a}|\mathbf{s},z) + \log q^\xi(z|\mathbf{s}) - \log \tilde{q}(z|\mathbf{a},\mathbf{s})\right)\right]$$
$$- D_{\mathrm{KL}}(\tilde{q}(z|\mathbf{a},\mathbf{s})\|q^\phi(z|\mathbf{s},\mathbf{a})) + const. \tag{36}$$

Since $D_{\mathrm{KL}}(\tilde{q}(z|\mathbf{a},\mathbf{s})\|q^\phi(z|\mathbf{s},\mathbf{a})) \geq 0$ we have the upper bound

$$U(\phi,\tilde{q}) = -\mathbb{E}_{\pi(\mathbf{a}|\mathbf{s})}\left[\sum_z \tilde{q}(z|\mathbf{a},\mathbf{s})\left(\log q^{\nu_z}(\mathbf{a}|\mathbf{s},z) + \log q^\xi(z|\mathbf{s}) - \log \tilde{q}(z|\mathbf{a},\mathbf{s})\right)\right]. \tag{37}$$

We can now write

$$U(\phi,\tilde{q}) = -\mathbb{E}_{\pi(\mathbf{a}|\mathbf{s})}\left[\sum_z \tilde{q}(z|\mathbf{a},\mathbf{s})\log q^{\nu_z}(\mathbf{a}|\mathbf{s},z)\right] + \mathbb{E}_{\pi(\mathbf{a}|\mathbf{s})}\left[D_{\mathrm{KL}}(\tilde{q}(z|\mathbf{a},\mathbf{s})\|q^\xi(z|\mathbf{s}))\right]. \tag{38}$$

Hence optimizing $U(\phi,\tilde{q})$ with respect to $\phi = \xi \cup \{\nu_z\}_z$ is equivalent to optimizing $D_{\mathrm{KL}}(\pi(\mathbf{a}|\mathbf{s})\|q^\phi(\mathbf{a}|\mathbf{s}))$. Specifically optimizing with respect to $\xi$ boils down to

$$\min_\xi U(\phi,\tilde{q}) = \min_\xi \mathbb{E}_{\pi(\mathbf{a}|\mathbf{s})}D_{\mathrm{KL}}(\tilde{q}(z|\mathbf{a},\mathbf{s})|q^\xi(z|\mathbf{s})) = \max_\xi \mathbb{E}_{\pi(\mathbf{a}|\mathbf{s})}\mathbb{E}_{\tilde{q}(z|\mathbf{a},\mathbf{s})}[\log q^\xi(z|\mathbf{s})]. \tag{39}$$

Noting that the expectation concerning $\mu(\mathbf{s})$ does not affect the minimizer, concludes the derivation.

## C   Environments

**Relay Kitchen**: A multi-task kitchen environment with long-horizon manipulation tasks such as moving kettle, open door, and turn on/off lights. The dataset consists of 566 human-collected trajectories with sequences of 4 executed skills. We used the same experiment settings and the pre-trained diffusion models from [9].

**XArm Block Push**: We used the adapted goal-conditioned variant from [9]. The Block-Push Environment consists of an XARm robot that must push two blocks, a red and a green one, into a red and green squared target area. The dataset consists of 1000 demonstrations collected by a deterministic controller with 4 possible goal configurations. The methods got 0.5 credit for every block pushed into one of the targets with a maximum score of 1.0. We use the pretrained Beso model from [9].

**D3IL** [13] is a simulation benchmark with diverse human demonstrations, which aims to evaluate imitation learning models' ability to capture multi-modal behaviors. D3IL provides 7 simulation tasks consisting of a 7DoF Franka Emika Panda robot and various objects, where each task has different solutions and the robot is required to acquire all behaviors. Except for success rate, D3IL proposes to use task behavior entropy to quantify the policy's capability of learning multi-modal distributions. Given the predefined behaviors $\beta$ for each task, the task behavior entropy is defined as,

$$\mathbb{E}_{s_0 \sim p(s_0)} \left[ \mathcal{H}\big(\pi(\beta|s_0)\big) \right] \approx -\frac{1}{S_0} \sum_{s_0 \sim p(s_0)} \sum_{\beta \in \mathcal{B}} \pi(\beta|s_0) \log_{|\mathcal{B}|} \pi(\beta|s_0) \qquad (40)$$

where $s_0$ refers to the initial state and $S_0$ refers to the number of samples from the initial state distribution $p(s_0)$. During the simulation, we rollout the policy multiple times for each $s_0$ and use a Monte Carlo estimation to compute the expectation of the behavior entropy.

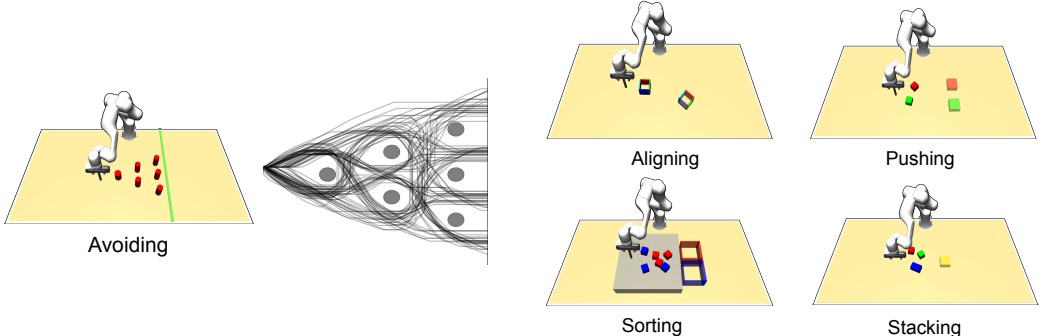

Figure 7: Visualization of D3IL tasks. We further provide the figure of demonstrations for the Avoiding task, which indicates 24 solutions of it.

In this paper, we evaluate our algorithm in Avoiding, Aligning, Pushing, and Stacking with state-based representations and Sorting and Stacking with image-based representations. The simulation environments can be found in Fig. 7. The **Avoiding** task requires the robot to reach the green line without colliding with any obstacles. The task contains 96 demonstrations, consisting of 24 solutions with 4 trajectories for each solution. The **Aligning** task requires the robot to push the box to match the target position and orientation. The robot can either push the box from inside or outside, thus resulting in 2 solutions. This task contains 1000 demonstrations, 500 for each solution with uniformly sampled initial states. The **Pushing** task requires the robot to push two blocks to the target areas. The robot can push the blocks in different orders and to different target areas, which gives the task 4 solutions. This task contains 2000 demonstrations, 500 for each solution with uniformly sampled initial states. The **Sorting** task requires the robot to sort red and blue blocks to the corresponding box. The number of solutions is determined by the sorting order. D3IL provides Sorting 2, 4, and 6 boxes, here we only use the Sorting-4 task, which contains around 1054 demonstrations with 18 solutions. The **Stacking** task requires the robot to stack three blocks in the target zone. Additionally, the blue block needs to be stacked upright which makes it more challenging. This task contains 1095 demonstrations with 6 solutions.

## D   Baselines Implementation

**BESO**   [9] is a continuous time diffusion policy that uses a continuous stochastic-differential equation to represent the denoising process. We implement this method from the D3IL benchmark, following the default which predicts one-step action conditional on the past five-step observations.

**DDPM**   [2, 10] is a discrete diffusion policy. We do not directly use the code from DiffusionPolicy [10] which implements an encoder-decoder transformer structure. For fair comparison, we evaluate this model from the D3IL benchmark which shares the same architecture as in BESO.

**Consistency Distillation**   [42] is designed to overcome the slow generation of diffusion models. Consistency models can directly map noise to data using one-step and few-step generation and they can be trained either through distilling pre-trained diffusion models or as an independent generative model. Our implementation takes the main training part of the model by integrating a GPT-based diffusion policy as the backbone.

**Consistency Trajectory Models** [44] is an extension of the CD model originally used in image generation. It augments the performance by integrating additional CTM and GAN loss terms in consideration. Later, it is used in robot policy prediction in [45] without taking the GAN loss. Our implementation of CTM is extended from our CD implementation, by modifying the loss computation.

**IMC** [14] is a curriculum-based approach that uses a curriculum to assign weights to the training data so that the policy can select samples to learn, aiming to address the mode-averaging problem in multimodal distributions. We implement the model using the official IMC code with a GPT structure.

**EM** [66] is based on the maximum likelihood objective and follows an iterative optimization scheme, where the algorithm switches between the M-step and the E-step in each iteration.

# E Hyper Parameters

## E.1 Hyperparameter Selection

We executed a large-scale grid search to fine-tune key hyperparameters for each baseline method. For other hyperparameters, we choose the value specified in their respective original papers. Below is a list summarizing the key hyperparameters that we swept during the experiment phase.

**BESO:** None

**DDPM:** None

**Consistency Distillation:** $\mu$: EMA decay rate, N: see Algorithm 2 in [42]. All the other hyperparameters reuse the ones from the diffusion policy (BESO), as CD requires to be initialized using a pre-trained diffusion model.

**Consistency Trajectory Models:** Same as Consistency Distillation.

**IMC-GPT:** Eta [14], Number of components

**EM-GPT:** Number of components

**VDD-DDPM:** Number of components, $t_{\min}$, $t_{\max}$

**VDD-BESO:** Number of components, $\sigma_{\min}$, $\sigma_{\max}$

| Methods / Parameters | Grid Search | Avoiding | Aligning | Pushing | Stacking | Sorting-Vision | Stacking-Vision | Kitchen | Block Push |
|---|---|---|---|---|---|---|---|---|---|
| **GPT (shared by all)** | | | | | | | | | |
| Number of Layers | – | 4 | 4 | 4 | 4 | 6 | 6 | 6 | 4 |
| Number of Attention Heads | – | 4 | 4 | 4 | 4 | 6 | 6 | 12 | 12 |
| Embedding Dimension | – | 72 | 72 | 72 | 72 | 120 | 120 | 240 | 192 |
| Window Size | – | 5 | 5 | 5 | 5 | 5 | 5 | 4 | 5 |
| Optimizer | – | Adam | Adam | Adam | Adam | Adam | Adam | Adam | Adam |
| Learning Rate $\times 10^{-4}$ | – | 1 | 1 | 1 | 1 | 1 | 1 | 1 | 1 |
| **DDPM** | | | | | | | | | |
| Number of Time Steps | – | 8 | 16 | 64 | 16 | 16 | 16 | | |
| **BESO** | | | | | | | | | |
| Number of Sampling Steps | – | 8 | 16 | 64 | 16 | 16 | 16 | | |
| $\sigma_{min}$ | – | 0.1 | 0.01 | 0.1 | 0.1 | 0.1 | 0.1 | 0.1 | 0.1 |
| $\sigma_{max}$ | – | 1 | 3 | 1 | 1 | 1 | 1 | 1 | 1 |
| **IMC-GPT** | | | | | | | | | |
| Number of Components | $\{4, 8, 20, 50\}$ | 8 | 8 | 8 | 8 | 8 | 8 | 20 | 20 |
| Eta $\eta$ | $\{0.5, 1, 2, 5\}$ | 1 | 1 | 1 | 1 | 2 | 5 | 1 | 1 |
| **EM-GPT** | | | | | | | | | |
| Number of Components | $\{4, 8, 20, 50\}$ | 20 | 20 | 50 | 20 | 20 | 20 | 20 | 20 |
| **CD** | | | | | | | | | |
| N, Algorithm 2 in [42] | $\{2, 5, 10, 20, 40, 80, 120, 180\}$ | 2 | 2 | 2 | 2 | 2 | 2 | 2 | 2 |
| EMA decay rate $\mu$ | $\{0.99, 0.999, 0.9999\}$ | 0.9999 | 0.9999 | 0.9999 | 0.9999 | 0.9999 | 0.9999 | 0.9999 | 0.9999 |
| **CTM** | | | | | | | | | |
| N, Algorithm 2 in [42] | $\{5, 10, 20, 40\}$ | 20 | 20 | 20 | 20 | 20 | 20 | 10 | 10 |
| EMA decay rate $\mu$ | $\{0.99, 0.999, 0.9999\}$ | 0.9999 | 0.9999 | 0.9999 | 0.9999 | 0.9999 | 0.9999 | 0.9999 | 0.9999 |
| **VDD-DDPM** | | | | | | | | | |
| Number of Components | – | 8 | 8 | 8 | 8 | 8 | 8 | 1 | 2 |
| $t_{\min}$ | $\{1, 2, 4\}$ | 1 | 4 | 4 | 2 | 1 | 6 | 0 | 0 |
| $t_{\max}$ | $\{4, 8, 10, 12\}$ | 4 | 6 | 12 | 8 | 4 | 12 | 3 | 3 |
| **VDD-BESO** | | | | | | | | | |
| Number of Components | $\{4, 8\}$ | 8 | 8 | 8 | 8 | 8 | 8 | 4 | 4 |
| $\sigma_{\min}$ | $\{0.1, 0.2, 0.4, 0.6, 0.8\}$ | 0.2 | 0.8 | 0.2 | 0.4 | 0.4 | 0.2 | 0.1 | 0.2 |
| $\sigma_{\max}$ | $\{0.1, 0.6, 0.8, 1.0\}$ | 0.5 | 1.0 | 0.5 | 1.0 | 0.8 | 0.5 | 0.1 | 0.6 |

Table 4: Hyperparameter algorithms proposed method and baselines. The 'Grid Serach' column indicates the values over which we performed a grid search. The values in the column which are marked with task names indicate which values were chosen for the reported results.

# F  Evaluation of Teacher Diffusion Models with Different Denoising Steps

Unlike DDPM, which uses a fixed set of timesteps, BESO learns a continuous-time representation of the scores. This continuous representation enables the use of various numerical integration schemes, which can impact the performance of the diffusion model. We conducted an evaluation on the BESO teacher we used with different denoising steps. The results are presented in Table 5.

| Environments | Euler Maru-16 | Euler Maru-32 | Euler Maru-64 | Euler Maru-16 | Euler Maru-32 | Euler Maru-64 |
|---|---|---|---|---|---|---|
| Avoiding | 0.96 | 0.96 | 0.95 | 0.87 | 0.86 | 0.88 |
| Aligning | 0.85 | 0.85 | 0.85 | 0.67 | 0.32 | 0.80 |
| Pushing | 0.77 | 0.80 | 0.78 | 0.71 | 0.78 | 0.76 |
| Stacking-1 | 0.91 | 0.87 | 0.88 | 0.30 | 0.32 | 0.32 |
| Stacking-2 | 0.70 | 0.64 | 0.63 | 0.13 | 0.14 | 0.13 |
| Sorting (image) | 0.70 | 0.76 | 0.76 | 0.19 | 0.23 | 0.24 |
| Stacking (image) | 0.60 | 0.70 | 0.70 | 0.15 | 0.16 | 0.21 |

Table 5: BESO with varies denoising steps.

## G   Compute Resources

We train and evaluate all the models based on our private clusters. Each node contains 4 NVIDIA A100 and we use one GPU for each method. We report the average training time in Table 6.

| | Training Time | |
| | state-based | image-based |
| --- | --- | --- |
| EM-GPT | $2-3$ h | $4-6$ h |
| IMC-GPT | $2-3$ h | $4-6$ h |
| VDD-DDPM | $3-4$ h | $6-8$ h |
| VDD-BESO | $3-4$ h | $6-8$ h |

Table 6: Training time for each method.

In addition, we evaluate how the number of trainable parameters and training time scale with different numbers of components. The results are presented in Table 7.

| Num. of Experts | 1 | 2 | 4 | 8 | 16 |
| --- | --- | --- | --- | --- | --- |
| Parameters ($\times 10^4$) | 144.23 | 144.45 | 144.89 | 145.76 | 147.50 |
| Times/1k Iters ($s$) | 64.48 | 70.19 | 74.62 | 106.98 | 166.54 |

Table 7: Adding more experts does not significantly increase the number of neural network parameters or the training time. We conducted the evaluation on the state-based avoiding task, using a machine with an RTX 3070 GPU and an i7-13700 CPU. This result is due to the optimized network architecture of the VDD model, as shown in Figure 5. Adding more experts will only increase the number of output linear layers, i.e., the mean and covariance nets, while the transformer backbone which contains most of the parameters remains unchanged.

