# OpenReview forum: "Variational Distillation of Diffusion Policies into Mixture of Experts"
_NeurIPS.cc/2024/Conference — NeurIPS 2024 poster_

### Official Review · Reviewer_5fPT · 2024-06-20

**Soundness:** 3
**Presentation:** 2
**Contribution:** 3
**Rating:** 6
**Confidence:** 4

**Summary:**

This paper introduces a theoretical method for extracting an MoE policy from a pretrained diffusion policy.  The advantage of MoE policy is that it allows faster sampling speed compared with the diffusion policy more stable training and better performance compared with methods that train MoE from scratch.

**Strengths:**

The proposed method solves the policy diversity issues in RL while maintaining high training/inference efficiency. The experiments show clear improvement over previous MoE-based imitation learning methods. The theories are well-proved and solid.

**Weaknesses:**

1. One main advantage of MoE policy against deterministic policy is its diversity. However, this paper seems to only focus on evaluating the final performance of algorithms and does not compare the diversity of different algorithms. Regarding on performance solely, it seems to me Gaussian policy + reverse KL objective is good enough, in what cases do we need more diverse policies? This needs to be justified.
2. Variantional distillation is a known technique in the 3D field. Although its application in RL is new and meaningful, this still indicates limited novelty. Could emphasize more on the new technique introduced in this paper when applying variational distillation for policy extraction.

**Questions:**

1. What is the mixture number n in your experiments?  Can you provide any experiment results ablating n? This could be very informative.
2. I highly suggest the authors provide the source code during rebuttal. Reproducibility is crucial for this kind of paper. (Even if it is just unsorted/raw code)

**Limitations:**

Above

---

> ### Author Rebuttal · Authors · 2024-08-07
>
> We would like to thank the reviewer for taking the time to review our work and the many helpful comments and suggestions. We hope the following replies sufficiently address the raised questions and concerns.  We will update the paper accordingly.
>
> ---
>
> > One main advantage of MoE policy against deterministic policy is its diversity. However, this paper seems to only focus on evaluating the final performance of algorithms and does not compare the diversity of different algorithms.
> >
>
> We thank the reviewer for sharing their concern. If we understand correctly, when the reviewer mentions "final performance" they are referring to the success rate. In our evaluation, we additionally assess the diversity of policies using *task entropy (TE)*.
>
> TE was originally introduced by [1] as a scalar metric ranging from 0 to 1, where 0 indicates that the model has only learned one way to solve the task, and 1 indicates that the model has learned all skills present in the demonstration data.
>
> We apologize for any confusion caused by our description of TE as a measure of versatility rather than diversity. It is indeed intended to quantify the model's ability to exhibit diverse behaviors. We will make the necessary changes to the experiment section to further enhance clarity.
>
> ---
>
> > Regarding on performance solely, it seems to me Gaussian policy + reverse KL objective is good enough, in what cases do we need more diverse policies? This needs to be justified.
> >
>
> We thank the reviewer for their suggestion to enhance the motivation for policies capable of learning diverse behavior. In response, we have incorporated additional points and references into the introduction of our paper.
>
> Additionally, we summarized these points below:
>
> 1. **Improving Generalization:** If the learned policy overfits a specific set of demonstrated behaviors, it may not generalize well to new situations. By exposing the model to diverse behaviors, the risk of overfitting is reduced, and the learned policy is more likely to capture the underlying principles of the task rather than memorizing specific trajectories [1, 2, 3].
> 2. **Enhancing Skill Transfer:** Learning diverse behaviors facilitates better skill transfer across different but related tasks. If the agent can imitate a wide range of behaviors, it is more likely to possess a set of skills that can be applied to various tasks, making it a more versatile and capable learner [1, 3].
> 3. **Behavior in complex multi-agent settings:** Following a multimodal policy through a diverse set of skills, has been shown to be advantageous in multi-agent settings, such as table tennis [4].
>
> ---
>
> > Variational distillation is a known technique in the 3D field. Although its application in RL is new and meaningful, this still indicates limited novelty. Could emphasize more on the new technique introduced in this paper when applying variational distillation for policy extraction.
> >
>
> We assume the reviewer is referring to [6] when mentioning that variational distillation is a known technique in the 3D field. We will add further details to the 'Related Works' section, highlighting the distinctions between their work and ours to emphasize the novelty introduced in our approach.
>
> In essence, [6] aims to generate 3D scenes using a pre-trained 2D teacher model for distillation. Their approach fundamentally differs from ours, as they focus on distilling knowledge from one diffusion model into another diffusion model. In contrast, our goal is to distill a diffusion model into another family of generative models, specifically Mixture of Experts, to achieve favorable properties such as faster inference and tractable likelihoods.
>
> ---
>
> > What is the mixture number n in your experiments?
> >
>
> To determine the value of n, we conducted a grid search and found that using n=8 yields consistently good performance across all D3IL tasks, and n=4 yields good results for Franka kitchen and block push. Additional details about the hyperparameters can be found in Appendix E.2, Table 4.
>
> We will also include more detailed information about the mixture number in the main section of the paper.
>
> ---
>
> > Can you provide any experiment results ablating n? This could be very informative.
> >
>
> We agree that such an ablation is very informative and in fact already included such this ablation in the original submission, see Figure 3a, with accompanying text in Section 5.5. The main takeaway is that increasing the mixture number improves task entropy and therefore the skill diversity of the learned policy. We will emphasize this ablation stronger in the next iteration of the paper.
>
> ---
>
> > I highly suggest the authors provide the source code during rebuttal. Reproducibility is crucial for this kind of paper. (Even if it is just unsorted/raw code)
> >
>
> We apologize for not including the source code in our initial submission. We have now provided a link to an anonymous GitHub repository containing the code for all our experiments. Additionally, we have included an IPython notebook to test our method on a small toy task, which essentially reproduces Figure 2. We have sent these links to the AC in an Official Comment, as the rebuttal guidelines prevent us from directly sharing these links in our response to the reviewers.
>
> ---
>
> We would like to thank the reviewer again and welcome the opportunity to address any additional concerns or questions that the reviewers may have.
>
> ---
> [1] Towards Diverse Behaviors: A Benchmark for Imitation Learning with Human Demonstrations, ICLR ‘2024
>
> [2] Neural Probabilistic Motor Primitives for Humanoid Control, ICLR ‘19
>
> [3] InfoGAIL: Interpretable Imitation Learning from Visual Demonstrations, NeurIPS ‘17
>
> [4] One Solution is Not All You Need: Few-Shot Extrapolation via Structured MaxEnt RL, NeurIPS ‘20
>
> [5] Specializing Versatile Skill Libraries using Local Mixture of Experts, CoRL ‘21
>
> [6] High-fidelity and diverse text-to-3d generation with variational score distillation, NeurIPS ‘23

---

> > ### Comment · Reviewer_5fPT · 2024-08-08
> > **Response**
> >
> > I thank the authors for the very detailed responses, which have resolved most of my concerns. I raise the score from 5 to 6.
> >
> > However, I would like to share one of my very personal opinions: Most (if not all) current public RL benchmarks and their evaluation metrics actually lack a strong need for policy diversity, which hinders the necessity of diversified policies. I believe designing new tasks and more difficult benchmarks is of critical importance for designing these algorithms.

---

> > > ### Author Response · Authors · 2024-08-08
> > > **We thank the reviewer for the positive feedback**
> > >
> > > We thank the reviewer for the positive feedback and are pleased that our response addressed their concerns. We fully agree that the community needs new benchmarks that require and evaluate diversity in RL settings. We are happy to address any other concerns that might arise during discussion session.

---

### Official Review · Reviewer_8iR5 · 2024-07-11

**Soundness:** 3
**Presentation:** 2
**Contribution:** 3
**Rating:** 5
**Confidence:** 4

**Summary:**

This study introduces Variational Diffusion Distillation (VDD), a novel method that distills pre-trained diffusion models into Mixture of Experts (MoE) frameworks. VDD addresses diffusion models' drawbacks of intractable likelihoods and long inference times, while leveraging their ability to represent complex distributions. By leveraging a decompositional variational objective, VDD trains MoEs efficiently, enabling real-time applications. VDD outperforms existing distillation and MoE training methods in several complex behavior learning tasks.

**Strengths:**

1) The technical approach appears to be sound, demonstrating a rigorous and well-grounded methodology.
2) The visualizations presented in the paper are meticulously crafted, showcasing a high level of attention to detail and aesthetics.
3) The experimental section is quite impressive, offering a substantial evaluation of the proposed method.

**Weaknesses:**

1) The challenges posed, although relevant to diffusion models and their potential difficulties in handling certain tasks or long inference times, lack sufficient evidence to conclusively state that these issues persist across all novel variants of diffusion models.
2) The use of identical phrasing in the abstract and introduction could potentially limit the reader's engagement, as it fails to provide a nuanced progression of ideas.
3) A missing period on line 218 detracts slightly from the overall readability and professionalism of the manuscript.
4) It appears that the experiments for the DDPM component in Table 1(a) are incomplete for the kitchen and block push datasets, which limits the comprehensiveness of the evaluation.
5) While the proposed method achieves commendable results, it does not consistently outperform all metrics, necessitating a deeper analysis to explain the observed variations and identify potential avenues for improvement.
6) The inconsistency in Table 3 left, where 2.16 is highlighted as better than 2.15 is confusing and suggests that the experiment may not have been fully completed.
7) The comparison to a limited number of methods may limit the ability to comprehensively assess the strengths and weaknesses of the proposed approach.
8) The lack of illustrative visualizations, such as good and bad case studies, in the main text hinders the reader's ability to fully grasp the performance and limitations of the method in real-world scenarios.

**Questions:**

see weaknesses

**Limitations:**

see weaknesses

---

> ### Author Rebuttal · Authors · 2024-08-07
>
> We would like to thank the reviewer for taking the time to review our work and the many helpful comments and suggestions. We hope the following replies sufficiently address the raised questions and concerns.  We will update the paper accordingly.
>
> > The challenges posed, although relevant to diffusion models and their potential difficulties in handling certain tasks or long inference times, lack sufficient evidence to conclusively state that these issues persist across all novel variants of diffusion models.
> >
>
> To the best of our knowledge, there are currently no variants of diffusion models that enable precise one-step generation with tractable likelihoods. Therefore, the study of diffusion distillation remains an ongoing area of investigation. Recent studies include [1, 2]. However, if the reviewer is aware of any novel diffusion model variants that address the challenges outlined in our work, we would be very happy to include them as references/baselines in our work.
>
> ---
>
> > The use of identical phrasing in the abstract and introduction could potentially limit the reader's engagement, as it fails to provide a nuanced progression of ideas
> >
>
> We thank the reviewer and will do our best to remove respective phrasing and make the paper more engaging for the reader.
>
> ---
>
> > […] missing period on line 218 […] The inconsistency in Table 3 left, where 2.16 is highlighted as better than 2.15 is confusing […]
> >
>
> We thank the reviewer for carefully reading the paper and identifying the formatting issues. We have addressed and corrected them.
>
> ---
>
> > It appears that the experiments for the DDPM component in Table 1(a) are incomplete for the kitchen and block push datasets, which limits the comprehensiveness of the evaluation.
> >
>
> We apologize for not including these results in the initial submission due to time constraints. We added the missing results to the Table R1 in PDF that accompanies the rebuttal.
>
> ---
>
> > While the proposed method achieves commendable results, it does not consistently outperform all metrics, necessitating a deeper analysis to explain the observed variations and identify potential avenues for improvement.
> >
>
> We thank the reviewer for sharing their concern. However, we want to emphasize that the primary goal of our paper is not to outperform all metrics, as the performance of our approach is limited by the diffusion ‘teacher’ model.
>
> Nevertheless, we agree with the reviewer that the submitted version of the paper did not sufficiently discuss future directions to enhance distillation capabilities. We will add this section to the manuscript and provide a summary below.
>
> **Future Work.** A promising avenue for further research is to utilize the features of the diffusion 'teacher' model to reduce training time and enhance performance. This can be achieved by leveraging the diffusion model as a backbone and fine-tuning an MoE head to predict the means and covariance matrices of the experts. The time-dependence of the diffusion model can be directly employed to train the MoE on multiple noise levels, effectively eliminating the need for the time-step selection scheme introduced in Section 4.3.
>
> ---
>
> > The comparison to a limited number of methods may limit the ability to comprehensively assess the strengths and weaknesses of the proposed approach.
> >
>
> To the best of our knowledge, we have considered the most recent and prominent baselines for training MoE models. However, we acknowledge that novel methods for distilling diffusion models, such as Consistency Trajectory Model (CTM) [3], were not included in our evaluation as they were not published at the time of submission.  In order to better evaluate the strength of VDD, we additionally add CTM as a new distillation baseline. The evaluation results are presented in the PDF that accompanies the rebuttal.
>
> ---
>
> > The lack of illustrative visualizations, such as good and bad case studies, in the main text hinders the reader's ability to fully grasp the performance and limitations of the method in real-world scenarios.
> >
>
> We thank the reviewer for bringing the need for illustrative visualizations to our attention. In response, we included additional visualizations (Figure R1) in the PDF that accompanies the rebuttal.
>
> Figure R1 highlights the most likely experts according to the gating probability at a given state. We can see that using a single component can achieve a perfect success rate at the cost of losing behavior diversity. Contrary, using many experts potentially results in a slightly lower success rate but increased behavior diversity. These qualitative results are consistent with the quantitative results from the ablation study presented in Figure 3a of the paper.
>
> ---
>
> We express our gratitude to the reviewer for their valuable comments and suggestions. We are pleased to address any additional questions or concerns that may arise.
>
> ---
>
> [1] Song, Yang, and Prafulla Dhariwal. "Improved techniques for training consistency models.” ICLR 2024
>
> [2] Xie, S., et al. “EM Distillation for One-step Diffusion Models”, Preprint
>
> [3] Kim, Dongjun, et al. "Consistency trajectory models: Learning probability flow ode trajectory of diffusion." ICLR 2024.

---

> > ### Comment · Area_Chair_Q8p8 · 2024-08-12
> >
> > Dear Reviewer,
> >
> > As the discussion period is nearing its conclusion, we kindly ask you to engage in the discussion and provide notes on any concerns that have not yet been addressed, along with the reasons why.
> >
> > Thank you for your attention to this matter.
> >
> > AC.

---

> > ### Comment · Reviewer_8iR5 · 2024-08-12
> >
> > I appreciate the authors for the efforts in answering my questions. Although some issues were not substantively addressed, I maintain my positive score due to the overall quality of the paper.

---

> > > ### Author Response · Authors · 2024-08-12
> > >
> > > We sincerely thank the reviewer for the positive comments on our work. We aimed to address all questions accurately and concisely, and we apologize if doing so left open questions. If the reviewer points us towards the respective issues, we would be happy to clarify and elaborate on them.

---

> > > ### Author Response · Authors · 2024-08-12
> > > **Regarding the unexpected lowering of the score**
> > >
> > > We are very surprised and saddened to see that the reviewer unexpectedly lowered their score without further explanation.
> > >
> > > We thoroughly checked the review and our rebuttal again and fail to see which concerns have not been substantively addressed.
> > >
> > > In particular, we want to emphasize that in response to the reviewer's concerns, we conducted additional experiments, evaluations and visualizations during the rebuttal period:
> > > - We performed DDPM distillation experiments on the Kitchen and Block Push datasets (in response to **Weakness 4**).
> > > - We introduced the recently proposed Consistency Trajectory Model (CTM) as a new baseline (in response to **Weakness 7**).
> > > - We included an illustrative visualization to provide deeper insights into our method (in response to **Weakness 8**).
> > >
> > > The remaining concerns have also been addressed in the original rebuttal:
> > > - **Weakness 1** was addressed and we offered to include additional discussion and baselines if pointed towards respective work.
> > >
> > > - **Weakness 2,3** and **6** were addressed by fixing typos and rephrasing sentences in the improved manuscript, which can not be uploaded as to rebuttal guidelines.
> > >
> > > - **Weakness 5** has been addressed and helped us clarify the goals, limitations and future work of our approach
> > >
> > > We sincerely hope there is an underlying misunderstanding and want to point out that the additional experiments and visualization are in the **PDF attached to the "Author Rebuttal" post** and not in the main manuscript (adhering to the rebuttal guidelines).
> > >
> > > We are of course happy to further clarify and elaborate on any of these concerns.

---

> ### Comment · Area_Chair_Q8p8 · 2024-08-10
> **Please reply to the rebuttal.**
>
> Dear Reviewer,
>
> Please reply to the rebuttal.
>
> AC.

---

### Official Review · Reviewer_MHge · 2024-07-12

**Soundness:** 3
**Presentation:** 3
**Contribution:** 3
**Rating:** 6
**Confidence:** 4

**Summary:**

This paper presents a variational inference method for distilling denoising diffusion policies into Mixture-of-Experts (MoE) policies. The primary motivation is to combine the strengths of both worlds - the ability to learn complex, multi-modal distributions of diffusion models - and the efficiency of MoEs offering faster inference and tractable likelihoods. Importantly, the decomposed upper bound of the variational objective enables separate, and thus more robust training for different experts. The proposed method is empirically evaluated on 9 behavior learning tasks. The author demonstrate its ability to distill complex distributions, outperforming existing distillation methods.

**Strengths:**

1. clear motivation of combining the strengths of both diffusion models and MoE.
2. combining decomposing objective with EM is shown to be effective to escape certain challenges of training MoE.
3. the paper is overall well-organized, with clear explanations of the method and detailed experimental results.

**Weaknesses:**

1. While EM is arguably be able to handle some limitations, it may also suffer from more longer convergence time during training. Moreover, while the authors discuss inference time improvements, this paper does not analyze training cost. As the training involves optimizing multiple experts and a gating - when the “optimal” number of experts is difficult to be pre-determined, it could potentially be computationally intensive. Is it possible to take task similarity into account and perform kind of meta-training like policy distillation?
2. The authors do not explicitly discuss how well VDD can capture long-range temporal dependency that might be presented in the original diffusion model, provided that it improves the inference speed. However, this can be important for sequential decision-making tasks.

**Questions:**

Please see Weaknesses section

**Limitations:**

The authors do have included discussion about the limitation and shed light on how they might be approaching in future studies.

---

> ### Author Rebuttal · Authors · 2024-08-07
>
> We would like to thank the reviewer for taking the time to review our work and the many helpful comments and suggestions. We hope the following replies sufficiently address the raised questions and concerns.  We will update the paper accordingly.
>
> ---
>
> > While EM is arguably be able to handle some limitations, it may also suffer from more longer convergence time during training. Moreover, while the authors discuss inference time improvements, this paper does not analyze training cost […] .
> >
>
> We thank the reviewer for bringing this issue to our attention. We apologize for not reporting the training times for a varying number of experts in the initial draft of the paper. We added a table showing the training costs using $n \in \{1,2,4,8,16\}$ components to the PDF that accompanies the rebuttal. Figure R2 shows that both, training time and the number of parameters increase sub-linearly with $n$.
>
> ---
>
> > As the training involves optimizing multiple experts and a gating - when the “optimal” number of experts is difficult to be pre-determined, it could potentially be computationally intensive.
> >
>
> Since the approach leverages variational inference, and hence the inherit mode seeking behavior of the reverse KL, VDD is relatively robust to the number of components. In fact, the results reported in the paper use $n=8$ for all D3IL tasks and $n=4$ for both kitchen and block push. Moreover, using a large number of components introduces only slight variations in performance criteria, as the gating network can deactivate those that are not needed to solve a task, as shown by the ablation study in Figure 3a in the paper and Figure R1 in the PDF that accompanies the rebuttal.
>
> ---
>
> > Is it possible to take task similarity into account and perform kind of meta-training like policy distillation?
> >
>
> We thank the reviewer for the interesting suggestion. We believe that using a shared feature embedding across tasks could potentially speed up training time and is a promising avenue for future research. Currently, we are exploring how to efficiently leverage the features learned by the teacher diffusion model to improve training times.
>
> ---
>
> > The authors do not explicitly discuss how well VDD can capture long-range temporal dependency that might be presented in the original diffusion model […]
> >
>
> VDD shares the same capacity for modeling temporal dependencies as the teacher diffusion model, as it employs the same transformer backbone. We argue that VDD's comparable success rates to the diffusion model demonstrate its ability to capture the long-range temporal dependencies inherent in the original diffusion model.
>
> ---
>
> We hope that our responses have addressed your concerns. We would be more than happy to address any remaining questions/concerns.

---

> > ### Comment · Reviewer_MHge · 2024-08-09
> >
> > I appreciate the authors for the efforts in answering my questions and I have no more questions at this stage. I will keep my score and lean towards positive outcome.

---

> > > ### Author Response · Authors · 2024-08-09
> > >
> > > We thank the reviewer for the positive evaluation and valuable suggestions, and we are pleased that our response addressed their concerns.  We are happy to address additional questions and suggestions to improve our work further.

---

### Official Review · Reviewer_YkuG · 2024-07-14

**Soundness:** 3
**Presentation:** 3
**Contribution:** 3
**Rating:** 6
**Confidence:** 4

**Summary:**

This paper presents Variational Diffusion Distillation (VDD), a method that distills denoising diffusion policies into Mixtures of Experts (MoE) using variational inference. Diffusion Models excel in learning complex distributions for behavior learning but have drawbacks like slow inference times. MoEs address these issues but are hard to train. In VDD, each expert can be trained separately under the corresponding guidance from the diffusion teacher. VDD demonstrates convincing performance in distilling complex distributions and outperforming existing methods across nine behavior learning tasks.

**Strengths:**

The paper addresses the problem of distilling an expressive but slow diffusion model into a faster Mixture of Experts (MoE) model. MoE is an ideal target for distillation due to its intrinsic structure and rapid computational process. Though closely related to Variational Score Distillation, the proposed extension to MoE generators is novel. The claims appear valid, and I found no issues in the derivations. Overall, the paper is well-written and clear.

**Weaknesses:**

- I like the idea of updating the gating of MoE, which has been proven to be crucial for the success of VDD in an ablation study. However, it appears a bit unnatural to me that the experts are learned with reverse KL with the gating is learned with forward KL, even though the minimizer for forward and reverse KLs should align. Authors may find it interesting to visit another perspective from a probably concurrent work (Xie et al., 2024), in which a new distillation framework is derived from forward KL.

- A very relevant work on distilling diffusion models, Luo et al. 2023, is missing.

- I observed some performance gaps between VDD-DDPM and VDD-BESO across various tasks even though the teachers perform similarly. I wonder if VDD is sensitive to the specific forward process used in the diffusion model.

- In my humble opinion, the author could try some more complex tasks to demonstrate the capabilities of VDD. For instance, implementing a fast and expressive model on a dexterous hand would benefit from an expressive policy like diffusion while also requiring speed. Prior work exists in directly learning an MoE image generation model with the EM algorithm from data (Yu et al., 2019). Given the power of diffusion teachers, I would expect a distillation method to perform better than directly learning from data.

Xie et al. 2024, EM Distillation for One-step Diffusion Models

Luo et al. 2023, Diff-Instruct: A Universal Approach for Transferring Knowledge From Pre-trained Diffusion Models

Yu et al. 2021, Unsupervised Foreground Extraction via Deep Region Competition

**Questions:**

- It appears that in simpler tasks like kitchen and block push, the distilled model underperforms compared to the original model. However, in more complex tasks, many distilled models seem to outperform the original. I am curious if the authors have some insights on this.

**Limitations:**

The authors provided the limitation statement, mainly concerning the expressity of MoE.

---

> ### Author Rebuttal · Authors · 2024-08-07
>
> We thank the reviewer for the very positive comments and are grateful for the valuable suggestions and comments.  We hope the following replies sufficiently address the raised questions and concerns.  We will update the paper accordingly.
>
> ---
>
> > Authors may find it interesting to visit another perspective from a probably concurrent work (Xie et al., 2024), in which a new distillation framework is derived from forward KL […] A very relevant work on distilling diffusion models, Luo et al. 2023, is missing.
> >
>
> We thank the reviewer for pointing us towards these very interesting and related works. We will include these references in the main manuscript and further highlight both the commonalities and differences between these works and our approach.
>
> ---
>
> > I observed some performance gaps between VDD-DDPM and VDD-BESO across various tasks even though the teachers perform similarly. I wonder if VDD is sensitive to the specific forward process used in the diffusion model.
> >
>
> Indeed, in our experience BESO appears to be distilled easier. We credit this to the different forward SDE which in the case of BESO is a simple (scaled) Wiener Process. In contrast, the DDPM SDE contains an additional non-zero drift term, which makes the distillation harder.
>
> ---
>
> > In my humble opinion, the author could try some more complex tasks to demonstrate the capabilities of VDD. For instance, implementing a fast and expressive model on a dexterous hand would benefit from an expressive policy like diffusion while also requiring speed […]
> >
>
> We thank the reviewer for the encouraging suggestion. However, we would like to emphasize that the tasks used in this work are taken from a very recent benchmark suite [1], allowing for the quantification of the behavior diversity of the learned policy. Maintaining this diversity through the distillation process is an integral part of our approach. Moreover, this task suite contains several complex manipulation tasks. These complexities are particularly evident in tasks such as *Sorting (Image)* and *Stacking (Image)*, where the teacher model achieves success rates of less than 70%.
>
> Nevertheless, we find the idea of incorporating the dexterous hand task very intriguing. In the future, we would like to further investigate this area and evaluate the performance of our model in this context.
>
> ---
>
> > It appears that in simpler tasks like kitchen and block push, the distilled model underperforms compared to the original model. However, in more complex tasks, many distilled models seem to outperform the original. I am curious if the authors have some insights on this.
> >
>
> We assume the reviewer refers to the success rate as a performance criterion when mentioning ‘outperform’. We acknowledge that it may seem confusing at first that the 'student' outperforms the 'teacher' in terms of success rate. In these cases we observed that student usually exhibits lower task entropy, indicating less diverse behavior.
>
> We hypothesize that there is a trade-off between mastering a single skill very well and learning multiple skills with slightly less accuracy. This hypothesis is further supported by the results of the ablation study presented in Figure 3a. Additionally, we have included further visual evidence supporting this hypothesis in the PDF file accompanying the rebuttal. Figure R1 shows that by reducing the number of experts the success rate increases, at the cost of losing skill diversity. The extreme case $Z=1$, has a success rate of 1, but 0 entropy.
> We add a discussion about this trade-off in the experiment section.
>
> ---
>
> We would like to thank the reviewers again for assessing our work. We would be delighted to address any additional questions or concerns they may have.
>
> ---
>
> [1] Towards Diverse Behaviors: A Benchmark for Imitation Learning with Human Demonstrations, ICLR ‘2024

---

> > ### Comment · Reviewer_YkuG · 2024-08-08
> >
> > Thank the authors for their response. I will keep my accepting rating.

---

> > > ### Author Response · Authors · 2024-08-09
> > >
> > > We sincerely appreciate the reviewer's positive evaluation and valuable suggestions. We are happy to address any additional questions and suggestions to further improve our work and its assessment.

---

### Official Review · Reviewer_14hC · 2024-07-15

**Soundness:** 3
**Presentation:** 3
**Contribution:** 3
**Rating:** 7
**Confidence:** 3

**Summary:**

This paper studies the knowledge distillation problem in diffusion models by distilling denoising diffusion policies into Mixtures of Experts (MoE) using variational inference. The goal is to combine the advantages of diffusion models, with the fast inference capabilities of MoE models. The authors construct an upper bound of a KL divergence-style loss function between the diffusion policy and the MoE policy and implement EM steps for variational inference to achieve the distillation. The proposed method, VDD (Variational Distillation of Diffusion), demonstrates superior performance compared to existing distillation and MoE methods.

**Strengths:**

1. The paper studies an interesting and practical problem in the distillation of diffusion models, achieving 1-step inference with performance comparable to the original diffusion policy. This type of fast inference can be highly beneficial for real-world control problems utilizing diffusion policies.
2. The proposed method, VDD, introduces a robust approach to distilling diffusion policies to  MoE models using VI.
3. Overall, the paper is well-structured, easy to understand, and provides a clear presentation of the proposed method.

**Weaknesses:**

1. In the limitations section, the paper claims that "VDD is not straightforwardly applicable to generating very high-dimensional data like images." However, the experimental results in Table 1(a) show that the distilled model VDD-DDPM outperforms the original DDPM diffusion policy in the Sorting (Image) and Stacking (Image) tasks, which contradicts the discussion. What is the reason for this? Why does the distilled model outperform the original diffusion policy?

2. Regarding training time, the time cost for distillation models (like VDD-DDPM) is higher than for MoE models. I assume this refers only to the knowledge distillation time. If we consider the original training time for diffusion policies, the total time cost may be significantly larger. It would be beneficial to report this total time for a fair comparison between MoE methods.

3. Does the distilled MoE model have the same architecture as the EM-GPT and IMC-GPT models?

4. While I agree that, according to this paper, the distillation of MoE achieves better performance than training from scratch, why do we distill from diffusion models? How about distilling from large and robust MoE models into smaller MoE models? This could be especially relevant in low-dimensional state observation cases, where MoE models reportedly perform better.

**Questions:**

1. Why does the distilled model outperform the original diffusion policy in Sorting (Image) and Stacking (Image) tasks?

2. Does the distilled MoE model have the same architecture as the EM-GPT and IMC-GPT models?

3. Why do we distill from diffusion models? How about distilling from large and robust MoE models into smaller MoE models?

**Limitations:**

The authors have discussed the limitations of the paper.

Please refer to the [weakness] section for detailed concerns and questions about the paper.

---

> ### Author Rebuttal · Authors · 2024-08-07
>
> We would like to thank the reviewer for taking the time to review our work and the many helpful comments and suggestions. We hope the following replies sufficiently address the raised questions and concerns.  We will update the paper accordingly.
>
> > […] the paper claims that "VDD is not straightforwardly applicable to generating very high-dimensional data like images."
> >
>
> The statement in the limitation section refers to the dimensionality of the action space. However, in the image-based version of sorting and stacking, the images represent the state space (model input). The model output remains a low-dimensional control signal for the robot. VDD is not straightforwardly applicable to tasks that require high-dimensional output space such as image generation, because predicting the Cholesky decomposition of the Covariance matrix scales quadratically with the dimensionality of output space. We will clarify the statement in the limitation section.
>
> ---
>
> > Why does the distilled model outperform the original diffusion policy in Sorting (Image) and Stacking (Image) tasks?
> >
>
> We assume the reviewer refers to the success rate as a performance criterion when mentioning ‘outperform’. We acknowledge that it may seem confusing at first that the 'student' outperforms the 'teacher' in terms of success rate. In these cases we observed that student usually exhibits lower task entropy, indicating less diverse behavior.
>
> We hypothesize that there is a trade-off between mastering a single skill very well and learning multiple skills with slightly less accuracy. This hypothesis is further supported by the results of the ablation study presented in Figure 3a. Additionally, we have included further visual evidence supporting this hypothesis in the PDF file accompanying the rebuttal. Figure R1 shows that by reducing the number of experts the success rate increases, at the cost of losing skill diversity. The extreme case $Z=1$, has a success rate of 1, but 0 entropy.
>
> We add a discussion about this trade-off in the experiment section.
>
> ---
>
> > Does the distilled MoE model have the same architecture as the EM-GPT and IMC-GPT models?
> >
>
> Yes, to ensure fairness across all experiments, we use the same architecture for these methods.
>
> ---
>
> > […] It would be beneficial to report this total time for a fair comparison between MoE methods.
> >
>
> We thank the reviewer for pointing this out. In the PDF file accompanying the rebuttal, we report the total training time as well as the separate training time of DDPM and VDD in Table R1(b).
>
> ---
>
> > Why do we distill from diffusion models? How about distilling from large and robust MoE models into smaller MoE models?
> >
>
> Diffusion models have shown great performance in recent studies, often making them the preferred choice for addressing complex tasks. However, they suffer from slow inference due the iterative denoising and do not offer a tractable likelihood. The goal of our work is, therefore, to mitigate these inherent downsides of diffusion models by distilling them into MoEs through variational inference. Nevertheless, we find the idea of leveraging our approach to distill large MoE into smaller ones very intriguing and will look into it in future work.
>
> ---
>
> We express our gratitude to the reviewer for their valuable comments and suggestions. We are pleased to address any additional questions or concerns that may arise.

---

> > ### Comment · Reviewer_14hC · 2024-08-09
> > **Response to the rebuttal**
> >
> > I appreciate the author's detailed responses, which have addressed most of my concerns.
> > However, there are still some aspects that remain unclear to me.
> >
> > 1. Why does the student VDD model outperform the teacher diffusion model?
> > The author suggests that this may be related to the trade-off between mastering a single skill very well and learning multiple skills with slightly less accuracy. However, when looking at the success rates in Table 1(a), the student model only outperforms in the Sorting (Image) and Stacking (Image) tasks, but not in others. Given a fixed teacher model and a fixed number of experts, it seems difficult to attribute this phenomenon to the acquisition of multiple skills or increased diversity. Could this be related to the use of image observations? Interestingly, the task entropy of VDD-BESO is comparable to BESO in the Sorting (Image) task. According to the paper, VDD-BESO is both more accurate and not less diverse than BESO. What could be the reason for this discrepancy?
> >
> > 2. How to choose the optimal number of experts in real-world control tasks?
> > The results indicate that reducing the number of experts can increase the success rate, albeit at the cost of skill diversity. While this conclusion seems reasonable, it raises the question: how should we determine the optimal number of experts in a Mixture of Experts (MoE) model for a given real-world control task or set of tasks? In many safety-critical control problems, success rate is often the primary concern, leading to the possibility of opting for a smaller number of experts. Could you provide guidance on this decision, or better yet, demonstrate through experiments any potential drawbacks of this strategy?

---

> > > ### Author Response · Authors · 2024-08-11
> > >
> > > > Why does the student VDD model outperform the teacher diffusion model? […] According to the paper, VDD-BESO is both more accurate and not less diverse than BESO. What could be the reason for this discrepancy?
> > > >
> > >
> > > We would like to express our sincere gratitude to the reviewer for their insightful questions, which prompted us to conduct further investigations. Below is a summary of our findings.
> > >
> > > Unlike DDPM, which uses a fixed set of timesteps, BESO learns a continuous-time representation of the scores. This continuous representation enables the use of various numerical integration schemes, which can impact the performance of the diffusion model.
> > >
> > > In our original experiments, we used the Euler-Maruyama method with 16 integration steps, following [1]. However, upon further investigation, we observed that utilizing alternative integration schemes, such as DDIM [2] or the Heun method [3], and increasing the number of integration steps, produced different success rates and entropy values. Some of these results surpassed those of VDD-BESO, underscoring that the teacher model indeed serves as an upper bound for the student model's performance. These findings are detailed in the table below. In the table we follow the convention "Sampler name - Number of integration steps”.
> > >
> > > |  | Euler_Maru-16 | Euler-16 | Heun-16 | DDIM-16 |
> > > | --- | --- | --- | --- | --- |
> > > | **Success Rate** | $0.70$ | $0.68$ | $0.72$ | $0.73$ |
> > > | **Task Entropy** | $0.19$ | $0.25$ | $0.25$ | $0.25$ |
> > > |  | **Euler_Maru-32** | **Euler_Maru-64** |  | **VDD-BESO** |
> > > | **Success Rate** | $0.76$ | $0.76$ |  | $0.76\pm0.04$ |
> > > | **Task Entropy** | $0.23$ | $0.24$ |  | $0.22\pm0.03$ |
> > >
> > > We, therefore, attribute VDD-BESO’s superior performance over BESO to the timestep selection strategy (see Section 4.3) of the learned continuous-time score functions.
> > >
> > > In response to the reviewer's comments, we will include the results of various integration schemes across all tasks and provide additional clarifications in our revised manuscript.
> > >
> > > ---
> > >
> > > > Could this be related to the use of image observations?
> > > >
> > >
> > > Regarding the performance gap observed primarily in image-based tasks, we suspect that this may be due to the use of the pre-trained vision encoder from the diffusion model for VDD, which likely enhances distillation performance. However, to confirm this hypothesis and provide a definitive explanation, further investigations are required.
> > >
> > > ---
> > >
> > > > How to choose the optimal number of experts in real-world control tasks?
> > > >
> > > >
> > > > The results indicate that reducing the number of experts can increase the success rate, albeit at the cost of skill diversity. While this conclusion seems reasonable, it raises the question: how should we determine the optimal number of experts in a Mixture of Experts (MoE) model for a given real-world control task or set of tasks? In many safety-critical control problems, success rate is often the primary concern, leading to the possibility of opting for a smaller number of experts. Could you provide guidance on this decision, or better yet, demonstrate through experiments any potential drawbacks of this strategy?
> > > >
> > >
> > > We understand that when the reviewer refers to "safety-critical control problems," they are addressing situations where failures or incorrect operations in control systems could lead to significant harm to humans or robots.
> > >
> > > To provide a meaningful response, we believe it is important to have more specific details about the particular task or setting.
> > >
> > > For example, in a static task environment where achieving high success rates is the primary objective, it might be advantageous to select a small number of components.
> > >
> > > On the other hand, if the task involves constantly changing dynamics, a policy that incorporates a diverse set of skills is likely to be more robust [4]. For instance, in the avoidance task depicted in Figure R1 of the PDF attached to our rebuttal, if the path learned by a single-expert policy (Z=1) becomes obstructed, the success rate would drop to zero. In such a dynamic environment, diverse policies could maintain some level of success in completing the task, making a larger number of experts the more suitable option.
> > >
> > > If our interpretation of "safety-critical control problems" differs from the reviewer’s definition, we would be happy to provide further clarification.
> > >
> > > ---
> > >
> > > [1] Jia X, Blessing D, Jiang X, Reuss M, Donat A, Lioutikov R, Neumann G. Towards diverse behaviors: A benchmark for imitation learning with human demonstrations. ICLR 2024.
> > >
> > > [2] Song J, Meng C, Ermon S. Denoising diffusion implicit models. ICLR 2021.
> > >
> > > [3] Karras T, Aittala M, Aila T, Laine S. Elucidating the design space of diffusion-based generative models. NIPS 2022.
> > >
> > > [4] Eysenbach, B. and Levine, S., 2021. Maximum entropy RL (provably) solves some robust RL problems. ICLR 2022.

---

> > > > ### Comment · Reviewer_14hC · 2024-08-13
> > > > **Response**
> > > >
> > > > Thank you for the author's response, which addressed most of my concerns and questions. I am raising my initial score from 6 to 7.

---

> > > > > ### Author Response · Authors · 2024-08-13
> > > > >
> > > > > We sincerely thank the reviewer for their positive feedback and constructive suggestions, which helped us improve the paper further. We are also pleased that our response addressed their concerns.

---

### Author Rebuttal · Authors · 2024-08-07

We thank all reviewers for their valuable feedback.  We would like to reiterate our main action points in response to the reviewers’ suggestions and concerns. New results can be found in the PDF file that accompanies the rebuttal.

- **Open-Sourced Code-Base:** We provided a link to an anonymous Github repository containing the code for reproducing the results of the paper. Additionally, we provided a link to an IPython notebook for testing our method on a simple toy task. We have sent both links to the AC in an Official Comment, as the rebuttal guidelines prevent us from sharing them here.
- **Added Baselines:** In response to feedback, we have included a comparison of our method with Consistency trajectory models [1].
- **Completed Results-Table.** The initial submission was missing the results for DDPM on two tasks, *kitchen* and *block push*. We have now included these results.
- **Detailed Parameter and Training Time Information:** Additional specifics on parameter counts and training times across varying number of experts have been incorporated.
- **Enhanced Visualizations:** To offer deeper insights into our method, we have included supplementary visualizations. It highlights the most likely experts according to the gating probability at a given state.

---

We would like to thank the reviewers again for assessing our work. We would be delighted to address any additional questions or concerns they may have during the upcoming discussion period.

---

[1] Kim D, Lai CH, Liao WH, Murata N, Takida Y, Uesaka T, He Y, Mitsufuji Y, Ermon S. Consistency trajectory models: Learning probability flow ode trajectory of diffusion. ICLR 2024.

---

### Author Response · Authors · 2024-08-14
**Thank you for the insightful discussions and help to improve VDD!**

Dear Reviewers,

We sincerely thank all of the reviewers for the thoughtful feedback and engaging discussions throughout the rebuttal process. Your insights have been invaluable in helping us improve our work on VDD. Going into the reviewer discussion, we would like to summarize the main improvements that resulted from your feedback during the rebuttal:

- **Related Work**: Expanded discussion on VDD's relation to recent diffusion distillation approaches to highlight our novelty.
- **Baseline Comparisons**: Added evaluations against Consistency Trajectory Model (CTM) to show the strong distillation performance of VDD.
- **Completed Results-Table:** Included results for DDPM and VDD-DDPM on kitchen and block push datasets.
- **Enhanced Evaluation on Teacher Models:** Added evaluations on diffusion teacher model with different samplers and different integration steps to show that VDD is able to maintain comparable performance with the teacher model.
- **Detailed Training Cost Information**: Provided clearer information on parameter counts and training times across varying number of experts. Reported the total training time of VDD including the training time of the diffusion teacher on kitchen and block push tasks.
- **Enhanced Visualization**: Added new visualization to offer better insights in to our method. It highlights the most likely experts according to the gating probability at a given state.
- **Open-Sourced Code-Base**: Provided a link to an anonymous Github repository containing the code for reproducing the results of the paper to the AC. Additionally, we provided a link to an IPython notebook for testing our method on a simple toy task.

These changes have significantly strengthened our paper and all discussed points will be included in the final version.

Thank you again for your time and valuable feedback and we hope you take these points into account for your final decision.

Sincerely,

The Authors

---

### Decision · Program_Chairs · 2024-09-25

**Decision:**

Accept (poster)

**Comment:**

This paper introduces Variational Diffusion Distillation (VDD), a method for distilling pre-trained diffusion policies into Mixture of Experts (MoE) models, combining the expressiveness of diffusion models with the fast inference of MoEs. VDD uses variational inference with a decomposed objective to train MoE experts efficiently. The method is evaluated on 9 behavior learning tasks, showing strong performance.

Reviewers appreciated the paper's technical approach, methodology, and extensive evaluation. Concerns included limited comparison to recent baselines and need for more analysis and visualizations. The authors addressed these in their rebuttal by adding comparisons, providing missing results, offering deeper analysis, and including additional visualizations. They also released code and provided details on hyperparameters and training costs.

Given the paper's strengths, thorough evaluation, and the authors' careful addressing of concerns, I recommend accepting this paper. The method enables practical use of expressive diffusion policies in domains requiring fast inference, representing a contribution to the field.